# Biliary NIK promotes ductular reaction and liver injury and fibrosis in mice

Zhiguo Zhang [1,7], Xiao Zhong[1,2,7], Hong Shen[1,7], Liang Sheng [1], Suthat Liangpunsakul[3], Anna S. Lok[4], M. Bishr Omary[1,5], Shaomeng Wang[6] & Liangyou Rui [1,4] ✉

Excessive cholangiocyte expansion (ductular reaction) promotes liver disease progression, but the underlying mechanism is poorly understood. Here we identify biliary NF-κB-inducing kinase (NIK) as a pivotal regulator of ductular reaction. NIK is known to activate the noncanonical IKKα/NF-κB2 pathway and regulate lymphoid tissue development. We find that cholangiocyte NIK is upregulated in mice with cholestasis induced by bile duct ligation (BDL), 5-diethoxycarbonyl-1,4-dihydrocollidine (DDC), or α-naphtyl-isothiocyanate (ANIT). DDC, ANIT, or BDL induces ductular reaction, liver injury, inflammation, and fibrosis in mice. Cholangiocyte-specific deletion of *NIK*, but not *IKKα*, blunts these pathological alterations. NIK inhibitor treatment similarly ameliorates DDC-induced ductular reaction, liver injury, and fibrosis. Biliary NIK directly increases cholangiocyte proliferation while suppressing cholangiocyte death, and it also promotes secretion of cholangiokines from cholangiocytes. Cholangiokines stimulate liver macrophages and hepatic stellate cells, augmenting liver inflammation and fibrosis. These results unveil a NIK/ductular reaction axis and a NIK/cholangiokine axis that promote liver disease progression.

Bile ducts provide conduits for bile flow from the liver to the gallbladder and along the journey, cholangiocytes–bile duct epithelial cells–modify bile compositions through absorption and/or secretion of various molecules. Biliary injury triggers cholangiocyte proliferation to compensate for bile duct loss and maintain biliary homeostasis. Bipotent cholangiocyte-like cells can differentiate into either cholangiocytes or hepatocytes, supporting liver regeneration[1]. However, excessive cholangiocyte expansion results in pathogenic ductular reaction, a hallmark of cholestasis, contributing to liver disease progression[2,3]. Ductular reaction marks poor prognosis and increases the risk for cholangiocarcinoma[3–6]. Bile duct ligation (BDL) induces

obstructive biliary injury to robustly stimulate ductular reaction and serves as a mouse ductular reaction model[7]. Exposure to biliary toxicant 3, 5-diethoxycarbonyl-1,4-dihydrocollidine (DDC) or α-naphtyl-isothiocyanate (ANIT) also profoundly induces ductular reaction in rodents, leading to cholestasis[8,9]. However, cholangiocyte-intrinsic pathways mediating ductular reaction remain poorly understood.

NF-κB-inducing kinase (NIK), also known as MAP3K14, is activated by a subset of cytokines and required for the activation of the noncanonical NF-κB2 pathway[10]. TNF-like weak inducer of apoptosis (TWEAK) is a NIK-stimulating cytokine and interestingly, TWEAK stimulates cholangiocyte proliferation via its receptor Fn14[11–13]. NIK

[1]Department of Molecular & Integrative Physiology, University of Michigan Medical School, Ann Arbor, MI 48109, USA. [2]Department of Infectious Diseases, Hunan Key Laboratory of Viral Hepatitis, Xiangya Hospital, Central South University, Changsha, China. [3]Division of Gastroenterology and Hepatology, Department of Medicine, Indiana University School of Medicine, Indianapolis, IN 46202, USA. [4]Division of Gastroenterology and Hepatology, Department of Internal Medicine, University of Michigan Medical School, Ann Arbor, MI 48109, USA. [5]Robert Wood Johnson Medical School, Center for Advanced Biotechnology and Medicine, Rutgers University, New Brunswick, NJ 08854, USA. [6]Departments of Internal Medicine, Pharmacology, and Medicinal Chemistry, University of Michigan Medical School, Ann Arbor, MI 48109, USA. [7]These authors contributed equally: Zhiguo Zhang, Xiao Zhong, Hong Shen. ✉e-mail: ruily@umich.edu

phosphorylates and activates IκB kinase-α (IKKα), also called Chuk[14–17]. IKKα in turn phosphorylates NF-κB2 precursor p100, resulting in proteolytic cleavage of p100 to generate mature transcription factor p52[18,19]. Additionally, NIK also activates the MAPK pathway and suppresses the JAK2/STAT3 pathway[20,21]. Functionally, lymphoid NIK critically regulates immune system development and immune response[10,22]. Global *NIK* knockout impairs lymph node and thymus developments[23]. Medullary thymic epithelial-specific *NIK* knockout abrogates mouse thymic medulla development, leading to fatal autoimmune hepatitis[24]. We reported that liver NIK is upregulated in obesity and chronic liver disease[15,23]. Hepatic NIK increases liver glucose production, hyperglycemia, and glucose intolerance in diabetes[14]. Aberrant activation of hepatic NIK also exacerbates hepatotoxin-induced liver failure in mice by impairing, at least in part, regenerative hepatocyte proliferation[21,25]. However, biliary NIK had not been explored prior to this study. Herein, we found that in mice and humans, NIK is robustly upregulated in cholangiocytes in cholestatic liver injury. Mice with cholangiocyte-specific *NIK* knockout (*NIK^ΔK19*) were resistant to DDC-, ANIT-, and BDL-induced ductular reaction, liver inflammation, and fibrosis. Likewise, NIK inhibitor treatment also mitigated DDC-induced ductular reaction, liver damage, and liver fibrosis. We demonstrated that biliary NIK critically regulates cholangiocyte secretion of soluble factors, termed cholangiokines, that stimulate liver macrophages and hepatic stellate cells (HSCs) and shape liver microenvironments. Our results unveil biliary NIK as a previously-unrecognized molecular driver for ductular reaction, liver injury, inflammation, and fibrosis, and raise the possibility that NIK inhibitors may have therapeutic potential for liver disease treatment.

## Results

### Biliary NIK is upregulated in mice and humans with chronic liver disease

To examine biliary NIK in humans, we coimmunostained liver sections with antibodies to NIK and cholangiocyte marker keratin-19 (K19). To validate anti-NIK antibody, we placed wild-type (*NIK^+/+*) mice on DDC diet for 2 weeks to increase liver NIK levels and used global *NIK* knockout (*NIK^−/−*) mice as negative control. Liver sections were stained with anti-NIK antibody. NIK-expressing (NIK^+) cells were readily detected in *NIK^+/+* mice but not in *NIK^−/−* mice (Supplementary Fig. 1A). In human samples, we observed a few NIK^+ cells and K19^+ cholangiocytes in healthy livers (Control) (Fig. 1A). The numbers of K19^+ cells, NIK^+ cells, and K19^+NIK^+ cells (NIK-expressing cholangiocytes) were considerably higher in primary biliary cholangitis (PBC), primary sclerosing cholangitis (PSC), hepatitis B virus (HBV), hepatitis C virus (HCV), and alcoholic cirrhosis (Fig. 1A, B). NIK-expressing cholangiocytes are likely to be underestimated because anti-NIK antibody may not detect NIK at a relatively low level.

To determine whether biliary NIK is upregulated in mice with cholestasis, we placed C57BL/6J mice on DDC diet for 4 weeks and stained liver sections with antibodies to NIK and K19. We observed a few NIK^+ cells and K19^+ cholangiocytes in chow-fed mice (Fig. 1C). DDC feeding dramatically increased the numbers of NIK^+ cells and K19^+ cholangiocytes; importantly, the majority of cholangiocytes expressed NIK (NIK^+K19^+) (Fig. 1C). It is worth mentioning that NIK^+K19^+ cholangiocytes are likely to be underestimated because anti-NIK antibody may not recognize NIK at relatively low levels. In line with these data, DDC feeding markedly increased liver *NIK* mRNA abundance (Fig. 1D). ANIT feeding for 3 weeks (Supplementary Fig. 1B, C) or BDL for 7 days (Fig. 1E, F) also markedly increased NIK^+K19^+ cholangiocyte number, liver *NIK* mRNA levels, and ductular reaction. Thus, biliary NIK upregulation is a signature of biliary injury regardless of etiology.

To test if cholangiocyte toxicants directly stimulate biliary NIK, we isolated primary cholangiocytes from *NIK^f/f* mice, immortalized them using E1A lentiviral vectors, and purified individual lines

through a series of dilutions. These lines expressed cholangiocyte-lineage genes, including *K19*, *Cftr*, and *Hnf1β* (Supplementary Fig. 2A, B), and were considered as cholangiocyte lines. We treated cholangiocytes with DDC or ANIT. Both DDC and ANIT significantly increased *NIK* mRNA abundance (Supplementary Fig. 2C). Endogenous NIK was below detection by anti-NIK antibody, so we transduced cholangiocytes with NIK adenoviral vectors. DDC or ANIT considerably increased NIK protein levels (Supplementary Fig. 2D). To assess NIK protein degradation, we pretreated cholangiocytes with DDC or ANIT and then treat them with protein translation inhibitor cycloheximide. DDC or ANIT pretreatment substantially increased NIK stability (Supplementary Fig. 2E, F). We loaded less lysates from DDC- or ANIT-treated cells (1/2 amounts of vehicle-treated cells) to avoid NIK overloading. To extend these observations, we examined the ubiquitin E3 ligase complex for NIK. Traf2 and Traf3 bind to NIK and recruit it to cIAP1 and cIAP2, and cIAP1/2 in turn catalyze NIK ubiquitination[10]. DDC treatment markedly decreased both cIAP1 and cIAP2, but increased Traf2 and Traf3, levels in cholangiocyte cultures (Supplementary Fig. 2G). To assess cIAP1/2 in vivo, we measured Traf2, Traf3, cIAP1, and cIAP2 levels in liver extracts by immunoblotting. DDC feeding markedly decreased liver cIAP1 and cIAP2 levels while increasing Traf2 and Traf3 levels (Supplementary Fig. 2H). To further confirm downregulation of hepatic cIAP1 and cIAP2, we immunoblotted liver extracts with antibodies to cIAP1 and cIAP2 from a different resource. Liver cIAP1 and cIAP2 were profoundly downregulated in DDC-treated mice (Supplementary Fig. 2I). ANIT feeding similarly decreased liver cIAP1/2 levels (Supplementary Fig. 2J). cIAP1/2 downregulation explains, at least in part, increased NIK stability and perhaps, upregulation of Traf2/3 is a secondary response to cIAP1/2 deficiency. Collectively, these results indicate that biliary injury increases both expression and stability of cholangiocyte NIK.

### Cholangiocyte-specific deletion of *NIK* suppresses DDC- and ANIT-induced ductular reaction

We sought to explore biliary NIK in vivo. We generated tamoxifen-induced, cholangiocyte-specific *NIK* knockout (*NIK^ΔK19*) mice by crossing *NIK^f/f* mice with *K19-CreERT* drivers. *NIK^f/f* and *K19-CreERT* mice have been described previously[26,27]. Because *NIK* and *K19-CreERT* genes reside in the same chromosome 11, *NIK^f/+;K19-CreERT^+/−* recombination occurs at a low frequency in germ cell meiosis. Nonetheless, we obtained *NIK^f/f;K19-CreERT^+/−* mice. *NIK^f/f;K19-CreERT^+/−* mice (7–10 weeks old) were treated with tamoxifen to delete *NIK* specifically in cholangiocytes, generating *NIK^ΔK19* mice. To verify cholangiocyte-specific deletion of *NIK*, we prepared primary cholangiocytes and hepatocytes from *NIK^f/f;K19-CreERT* mice, treated them with tamoxifen, and measured the disrupted *NIK* allele (*NIK^−/−*) by PCR-based genotyping. The *NIK^−/−* allele was detected in tamoxifen-treated cholangiocytes, but not in tamoxifen-treated hepatocytes or vehicle-treated cholangiocytes (Supplementary Fig. 3A). We placed *NIK^ΔK19* and control mice (2 weeks post tamoxifen treatment) on DDC diet for additional 4 weeks. We used 2 control groups. *NIK^f/f;K19-CreERT^+/−* mice were treated with olive oil vehicles as control (hereafter referred to as *NIK^f/f*). As an additional control, *NIK^f/f* mice were treated with tamoxifen for 2 weeks followed by a DDC feeding for 4 weeks. DDC-induced liver injury to a comparable level between tamoxifen-treated and olive oil vehicle-treated *NIK^f/f* mice (Supplementary Fig. 3B), indicating that tamoxifen treatment alone did not affect the response of the liver to DDC. Given that tamoxifen-treated *NIK^f/f* mice were indistinguishable from vehicle-treated *NIK^f/f;K19-CreERT* mice, they were not included in the following experiments.

DDC feeding decreased body weight in both *NIK^ΔK19* and *NIK^f/f* littermates, but to a higher level in *NIK^f/f* mice (Supplementary Fig. 3C). To grossly estimate biliary growth, we dissected intrahepatic bile ducts using collagenase. DDC feeding increased total bile duct weight in both *NIK^ΔK19* and *NIK^f/f* littermates, but to a markedly higher level in *NIK^f/f*

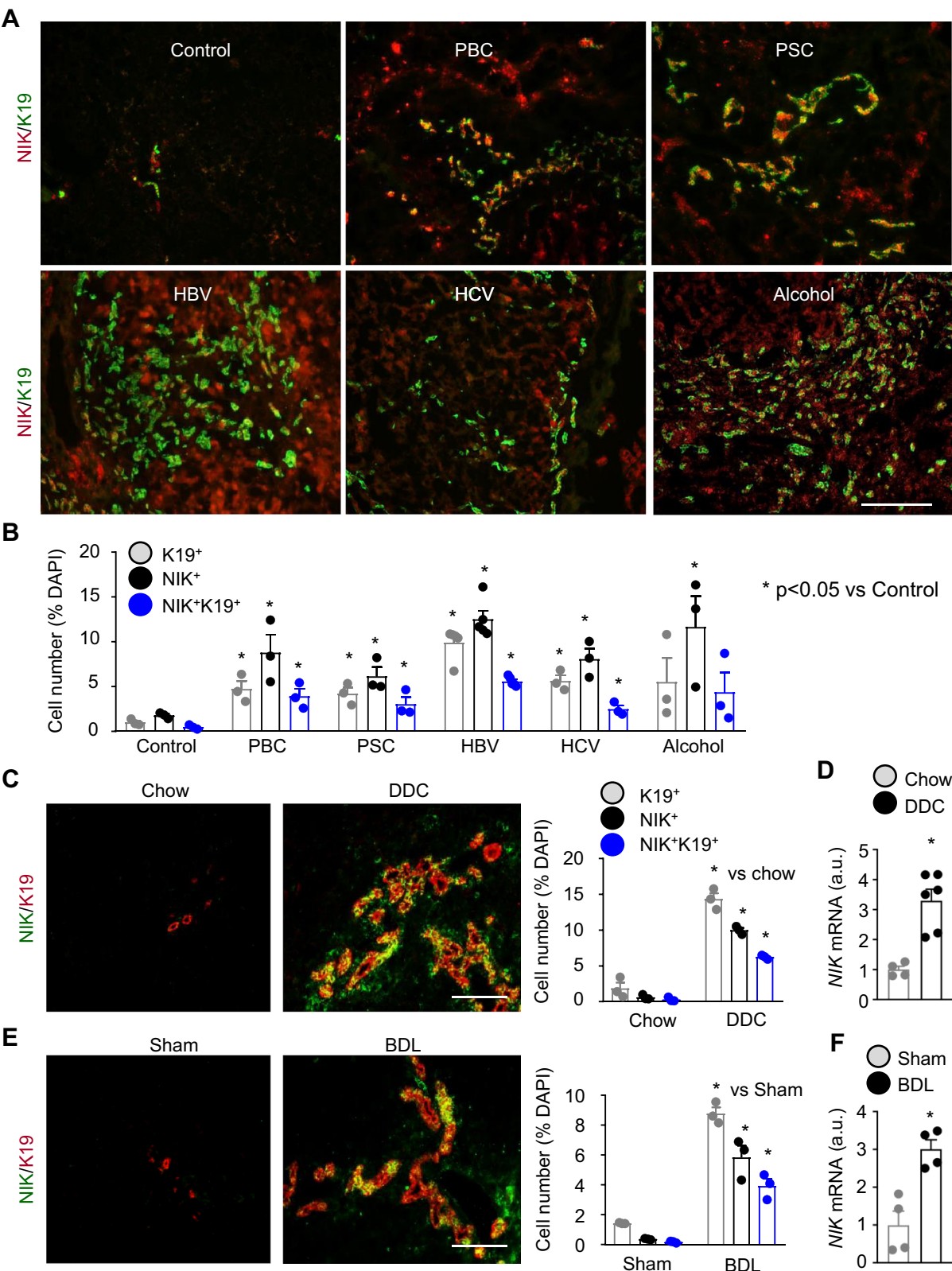

mice (Supplementary Fig. 3D). To directly examine ductular reaction, we co-stained liver sections with antibodies to K19 and NIK. As expected, DDC feeding induced dramatic ductular reaction and expansion of NIK+K19+ cholangiocytes in wild-type *NIK^{f/f}* mice (Fig. 2A, B). K19+ cholangiocyte number was drastically reduced in *NIK^{ΔK19}* mice (Fig. 2A). Liver NIK+K19+ cells were undetectable in *NIK^{ΔK19}* mice (Fig. 2A, B), confirming that *NIK* was deleted specifically in cholangiocytes. To

further verify *NIK* deletion in *NIK^{ΔK19}* mice, we immunoblotted liver extracts with anti-NF-κB2 antibody. DDC feeding markedly increased NF-κB2 p100 levels and p100 conversion into p52 (NIK activation index) in *NIK^{f/f}* mice, and p52 levels were dramatically lower in *NIK^{ΔK19}* mice (Fig. 2C). To confirm that deficiency of biliary NIK blocks DDC-induced ductular reaction, we immunostained liver sections with anti-K19 antibody. We observed a few K19+ cholangiocytes in chow-fed

**Fig. 1 | Chronic liver disease is associated with NIK upregulation in cholangiocytes. A, B** Human liver sections were stained with antibodies to NIK and K19. **A** Representative images. Scale bar: 200 μm. **B** NIK⁺, K19⁺, and NIK⁺K19⁺ cells were counted and normalized to total cells. Control: $n = 3$ subjects, PBC: $n = 3$ subjects, PSC: $n = 3$ subjects, HBV: $n = 5$ subjects, HCV: $n = 3$ subjects, Alcohol: $n = 3$ subjects. **C, D** C57BL/6J male mice were fed a chow or DDC diet for 4 weeks. **C** Liver sections were stained with antibodies to NIK and K19. NIK⁺, K19⁺, and NIK⁺K19⁺ cells were counted and normalized to total cells. Chow: $n = 3$ mice, DDC: $n = 3$

mice. Scale bar: 200 μm. **D** Liver NIK expression was measured by qPCR (normalized to 18 S levels). Chow: $n = 4$ mice, DDC: $n = 6$ mice. a.u. arbitrary units. **E, F** C57BL/6J males were treated with BDL or sham surgery for 7 days. **E** Liver sections were stained with antibodies to NIK and K19. NIK⁺, K19⁺, and NIK⁺K19⁺ cells were counted and normalized to total cells ($n = 3$ mice per group). Scale bar: 200 μm. **F** Liver NIK expression was measured by qPCR (normalized to 36B4 levels, $n = 4$ mice per group). Data are presented as mean ± SEM. *$p < 0.05$, 2-tailed student's $t$-test. Source data are provided as a Source Data file.

*NIK^{f/f}* and *NIK^{ΔK19}* mice; importantly, DDC feeding increased K19⁺ cholangiocyte number to a dramatically higher level in *NIK^{f/f}* mice relative to *NIK^{ΔK19}* mice (Fig. 2D, E). Liver *K19* mRNA levels were also significantly higher in *NIK^{f/f}* than in *NIK^{ΔK19}* mice (Fig. 2F). NIK⁺HNF4α⁺ hepatocytes were barely detected in chow-fed mice and modestly increased after DDC feeding (Supplementary Fig. 3E, F). Unlike NIK⁺K19⁺ cholangiocytes, NIK⁺HNF4α⁺ hepatocytes were comparable between *NIK^{f/f}* and *NIK^{ΔK19}* mice (Supplementary Fig. 3F). To extend these findings, we placed *NIK^{ΔK19}* and *NIK^{f/f}* littermates on an ANIT diet for 3 weeks. ANIT, like DDC, induced a marked cholangiocyte expansion in *NIK^{f/f}* mice; importantly ANIT-induced ductular reaction was also dramatically attenuated in *NIK^{ΔK19}* mice compared to *NIK^{f/f}* mice (Fig. 2G and Supplementary Fig. 3G). These results unveil NIK as an indispensable cholangiocyte-intrinsic inducer of ductular reaction.

## Biliary NIK promotes cholangiocyte proliferation while suppressing cholangiocyte death

We set out to interrogate the cellular mechanism of NIK-induced ductular reaction. We placed *NIK^{f/f}* and *NIK^{ΔK19}* mice on DDC diet for 4 weeks. Liver sections were coimmunostained with antibodies to K19 and Ki67 (proliferation marker) or K19 and cleaved caspase-3 (apoptosis marker). The number of proliferating Ki67⁺K19⁺ cholangiocytes was significantly lower in *NIK^{ΔK19}* than in *NIK^{f/f}* mice; in contrast, the number of apoptotic caspase-3⁺K19⁺ cholangiocytes was significantly higher in *NIK^{ΔK19}* mice (Fig. 3A). Using TUNEL assays, we confirmed that cholangiocyte death was higher in *NIK^{ΔK19}* mice relative to *NIK^{f/f}* littermates post DDC diet (Supplementary Fig. 4A). Of note, proliferation of HNF4α⁺ hepatocytes and F4/80⁺ Kupffer cells, as well as hepatocyte death, were comparable between *NIK^{ΔK19}* and *NIK^{f/f}* mice (Supplementary Fig. 4A). We also examined cholangiocyte proliferation and death in ANIT-fed mice (3 weeks). Cholangiocyte proliferation (Ki67⁺K19⁺) was significantly lower, whereas cholangiocytes death (caspase-3⁺K19⁺) was significantly higher, in *NIK^{ΔK19}* mice than in *NIK^{f/f}* mice (Fig. 3B).

To test if NIK cell-autonomously regulates cholangiocyte proliferation and survival, we generated *NIK*-deficient cholangiocyte lines. We prepared cholangiocyte lines from *NIK^{f/f}* mice (Supplementary Fig. 2A, B). *NIK^{f/f}* cholangiocytes (hereafter referred to as *NIK^{+/+}*) were transduced with Cre adenoviral vectors to delete *NIK* (*NIK^{−/−}*), and adenoviral vectors were subsequently removed by repeatedly passaging. To confirm *NIK* deletion, we stimulated *NIK^{+/+}* and *NIK^{−/−}* cholangiocytes with TWEAK. TWEAK-stimulated NF-κB2 p52 production in *NIK^{+/+}* but not *NIK^{−/−}* cholangiocytes (Supplementary Fig. 5A). To examine proliferation, we stimulated *NIK^{+/+}* and *NIK^{−/−}* cholangiocytes with TWEAK or fetal bovine serum (FBS) and counted cell numbers. TWEAK increased *NIK^{+/+}* but not *NIK^{−/−}* cholangiocyte number (Fig. 3C). FBS increased *NIK^{+/+}* cholangiocytes to a higher level relative to *NIK^{−/−}* cholangiocytes (Fig. 3C). We verified these results using BrdU incorporation assays. BrdU-positive cells were significantly higher in *NIK^{+/+}* than in *NIK^{−/−}* cholangiocyte cultures post TWEAK or FBS stimulation (Fig. 3D and Supplementary Fig. 5B). To induce apoptosis, cholangiocytes were deprived of serum for 24 h or treated with palmitate, and cell death was measured using TUNEL assays. Palmitate was reported to induce cholangiocyte death[28]. Serum starvation or palmitate treatment increased TUNEL⁺ cells to a significantly higher level in *NIK^{−/−}* than in *NIK^{+/+}* cholangiocyte cultures (Fig. 3E and Supplementary Fig. 5C). Serum deprivation reduced cell viability to a higher level in *NIK^{−/−}* than

in *NIK^{+/+}* cholangiocyte cultures, as assessed by MTT assays (Supplementary Fig. 5D). These results suggest that biliary NIK promotes ductular reaction by enhancing cholangiocyte proliferation and suppressing its death.

## Ablation of biliary NIK alleviates DDC- and ANIT-induced liver injury, inflammation, and fibrosis

We next sought to investigate the pathological consequence of ductular reaction. Body weight, plasma alanine aminotransferase (ALT), alkaline phosphatase (ALP), and bilirubin were normal in *NIK^{ΔK19}* mice on chow diet (Supplementary Fig. 6A, B). We placed *NIK^{ΔK19}* and *NIK^{f/f}* littermates on a DDC diet for 4 weeks. Plasma ALT increased within a week post DDC diet and declined afterwards, and plasma ALP and bilirubin levels (biliary injury markers) increased progressively on DDC diet (Fig. 4A). Importantly, plasma ALT, ALP, and bilirubin levels were significantly lower in *NIK^{ΔK19}* than in *NIK^{f/f}* mice post DDC diet (Fig. 4A). DDC feeding induced severe liver damage and inflammation as detected by histological examinations; strikingly, liver damage and immune cell infiltration were substantially lower in *NIK^{ΔK19}* than in *NIK^{f/f}* mice (Fig. 4B). Liver CD11b⁺ macrophages and Gr-1⁺ myeloid cells were significantly less in *NIK^{ΔK19}* than in *NIK^{f/f}* mice (Fig. 4B, C). NIK⁺CD11b⁺ cells were comparable between *NIK^{ΔK19}* and *NIK^{f/f}* mice on DDC diet (Supplementary Fig. 3F). Liver CD4 and CD8 T cells were slightly lower in *NIK^{ΔK19}* mice (Supplementary Fig. 4B). To corroborate these results, we placed *NIK^{ΔK19}* and *NIK^{f/f}* littermates on an ANIT diet for 3 weeks. ANIT feeding increased plasma ALT, ALP, and bilirubin to a significantly higher level in *NIK^{f/f}* mice than in *NIK^{ΔK19}* mice (Supplementary Fig. 7A). *NIK^{ΔK19}* mice were also markedly resistant to ANIT-induced liver damage and inflammation (Fig. 4D, E and Supplementary Fig. 7B).

We examined liver fibrosis by Sirius red staining of liver sections and measuring liver hydroxyproline content, and we also examined α-smooth muscle actin (αSMA)-expressing HSCs that are responsible for liver fibrosis. Sirius red-positive fibers and αSMA⁺ HSCs were barely detectable in chow-fed mice and were robustly increased after DDC feeding (Fig. 5A). Sirius red areas, hydroxyproline content, and αSMA⁺ HSC number were substantially lower in *NIK^{ΔK19}* than in *NIK^{f/f}* mice on DDC diet (Fig. 5A, B). Liver extract αSMA levels were markedly lower in *NIK^{ΔK19}* than in *NIK^{f/f}* mice post DDC feeding (Fig. 5C). To determine whether sex influences the phenotypes, we placed *NIK^{ΔK19}* and *NIK^{f/f}* female mice on a DDC diet for 4 weeks. *NIK^{ΔK19}* females, like males, were resistant to DDC-induced ductular reaction, liver injury, inflammation, and fibrosis (Supplementary Fig. 8). To further validate these findings, we placed *NIK^{f/f}* and *NIK^{ΔK19}* mice on an ANIT diet for 3 weeks. ANIT robustly stimulated αSMA⁺ HSC activation and liver fibrosis in *NIK^{f/f}* mice; HSC activation and liver fibrosis were largely blocked in *NIK^{ΔK19}* mice (Fig. 5D). These results demonstrate, for the first time, that cholangiocyte-intrinsic NIK promotes liver damage, inflammation, and fibrosis, perhaps through inducing ductular reaction.

## *NIK^{ΔK19}* mice are resistant to BDL-induced ductular reaction and liver injury

To test if biliary NIK provides a common route to ductular reaction and liver disease, we performed BDL for 7 days on *NIK^{ΔK19}* and *NIK^{f/f}* mice. Plasma ALT, ALP, and bilirubin levels were comparable between sham-treated *NIK^{ΔK19}* and *NIK^{f/f}* mice (Supplementary Fig. 9A). BDL increased

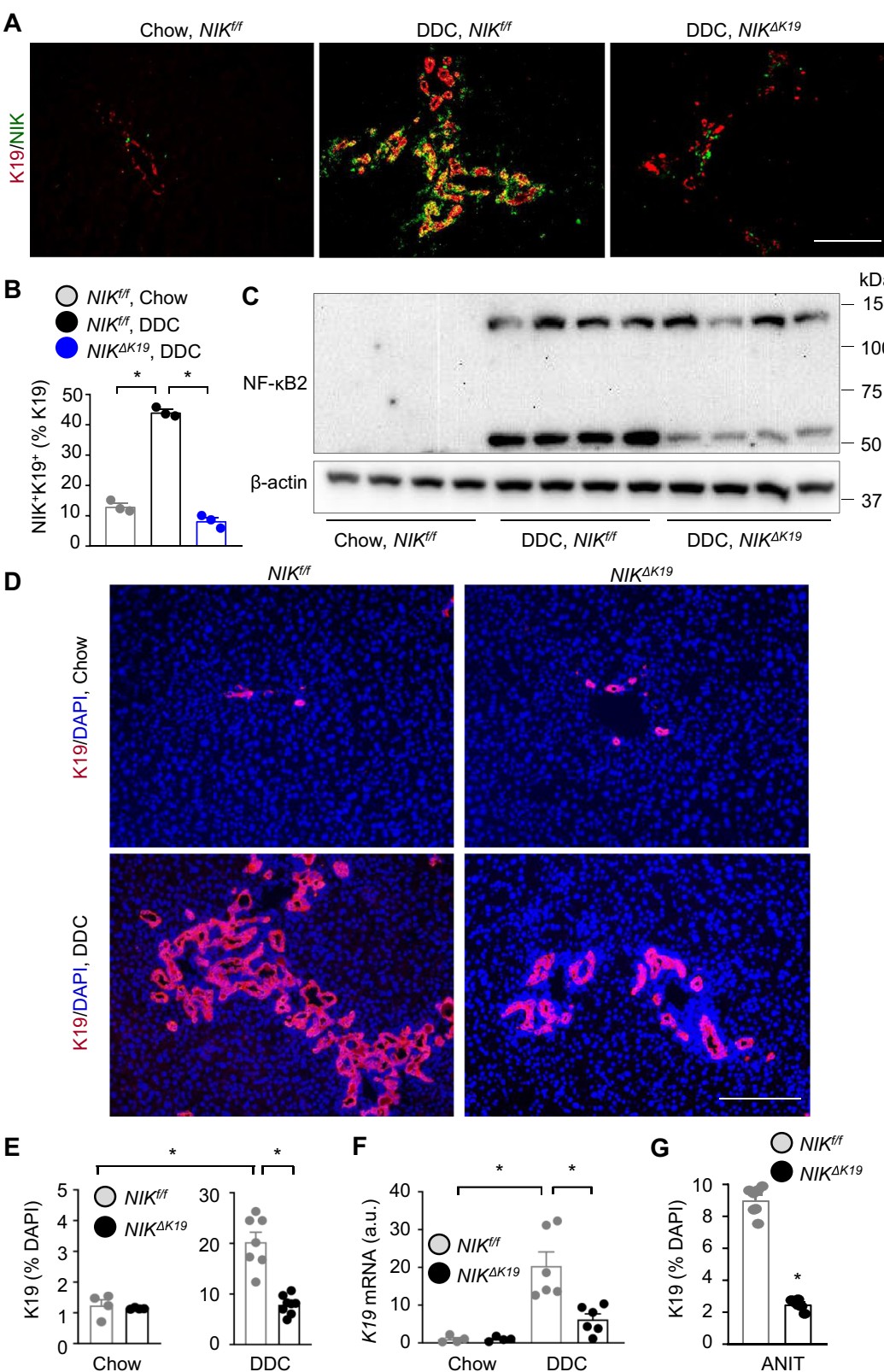

plasma ALT, ALP, and bilirubin levels to a significantly higher extent in *NIK^f/f* than in *NIK^ΔK19* littermates (Fig. 6A). BDL-induced liver injury, ductular reaction (K19 staining), liver inflammation (H&E, CD11b, and Gr-1 staining), and fibrosis (Sirius red and αSMA staining) in *NIK^f/f* mice (Fig. 6B), and *NIK^ΔK19* mice were resistant to BDL-induced liver damage, ductular reaction, inflammation, and fibrosis (Fig. 6B, C).

Cholangiocyte proliferation (K19⁺Ki67⁺) was lower while cholangioyte apoptosis (K19⁺caspase-3⁺) was higher in *NIK^ΔK19* mice relative to *NIK^f/f* mice post BDL (Fig. 6B and Supplementary Fig. 9B). Collectively, these findings suggest that biliary NIK integrates, as a converging point, information from diverse biliary insults to induces ductular reaction, liver injury, inflammation, and fibrosis.

**Fig. 2 | Ablation of biliary NIK attenuates DDC-induced ductular reaction.**
A–F $NIK^{\Delta K19}$ and $NIK^{f/f}$ male mice were fed a chow or DDC diet for 4 weeks. **A, B** Liver sections were costained with antibodies to NIK and K19. NIK⁺K19⁺ cells were counted and normalized to K19⁺ cells (n = 3 mice per group). Scale bar: 200 µm. **C** Liver extracts were immunoblotted with antibodies to NF-κB2 and β-actin (each lane represents an individual mouse; 4 mice per group). **D, E** Liver sections were immunostained with anti-K19 antibody. **D** Representative images. **E** K19 cholangiocytes were counted and normalized to total cell number. Chow: $NIK^{f/f}$: n = 4

mice, $NIK^{\Delta K19}$: n = 4 mice; DDC: $NIK^{f/f}$: n = 7 mice, $NIK^{\Delta K19}$: n = 8 mice. Scale bar: 200 µm. **F** Liver K19 expression (normalized to 18 S levels). a.u. arbitrary units. Chow: $NIK^{f/f}$: n = 4 mice, $NIK^{\Delta K19}$: n = 4 mice; DDC: $NIK^{f/f}$: n = 6 mice, $NIK^{\Delta K19}$: n = 6 mice. **G** $NIK^{\Delta K19}$ (n = 6 mice) and $NIK^{f/f}$ (n = 7 mice) males were fed an ANIT diet for 3 weeks. Liver sections were stained with anti-K19 antibody. K19 cholangiocytes were counted and normalized to total liver cells. Data are presented as mean ± SEM. *$p < 0.05$, 2-way ANOVA **B, E, F** and 2-tailed student's $t$-test **G**. Source data are provided as a Source Data file.

## Biliary NIK promotes secretion of pro-inflammation and pro-fibrosis cholangiokines

We next set out to investigate the mechanism by which biliary NIK-elicited ductular reaction promotes liver inflammation and fibrosis. We postulated that NIK might stimulate secretion of cholangiokines (cholangiocyte-derived mediators) that activate liver immune cells and HSCs, augmenting liver inflammation and fibrosis. We stimulated $NIK^{+/+}$ and $NIK^{-/-}$ cholangiocyte cultures with TWEAK to activate NIK and measured expression of various cytokines known to be involved in liver inflammation and fibrosis. Ablation of NIK significantly decreased expression of Il-1β, Il-4, Il-6, iNos, Tnfα, Mcp1 (also called Ccl2), and Tgfβ1 in $NIK^{-/-}$ cholangiocytes relative to $NIK^{+/+}$ cholangiocytes (Fig. 7A). To test if NIK-upregulated cholangiokines promote liver inflammation, we isolated mouse bone marrow-derived macrophages (BMDMs) and cocultured them with either $NIK^{+/+}$ or $NIK^{-/-}$ cholangiocytes in transwells as we described previously[15]. Cholangiocytes were pretreated with TWEAK to activate NIK. BMDM activity was assessed by measuring expression of inflammatory mediators. Expression of Il-1β, Il-6, iNos, Tnfα, Mcp1, and Ccl5 was significantly lower in $NIK^{-/-}$ cholangiocyte-cocultured BMDMs relative to $NIK^{+/+}$ cholangiocyte-cocultured BMDMs (Fig. 7B). To further verify that biliary NIK stimulates secretion of cholangiokines that activate BMDMs, we stimulated $NIK^{+/+}$ and $NIK^{-/-}$ cholangiocytes with TWEAK to increase secretion of cholangiokines into growth medium and prepared conditioned medium from these cells. BMDMs were treated with $NIK^{+/+}$ or $NIK^{-/-}$ cholangiocyte-conditioned medium and their expression of inflammatory mediators was measured. Expression of Il-1β, Il-6, iNos, Tnfα, Mcp1, and Ccl5 was significantly lower in $NIK^{-/-}$ conditioned medium-treated BMDMs than in $NIK^{+/+}$ conditioned medium-treated BMDMs (Fig. 7C). To determine whether NIK-upregulated cholangiokines stimulate HSCs and liver fibrosis, we treated HSCs with $NIK^{+/+}$ or $NIK^{-/-}$ cholangiocyte-conditioned medium and measured their expression of fibrosis-related genes. Expression of αSMA, Col1a1, Timp1, Mmp9, Ctgf, and Tgfβ1 was significantly lower in $NIK^{-/-}$ than in $NIK^{+/+}$ conditioned medium-stimulated HSCs (Fig. 7D). These results provide proof of concept evidence that biliary NIK potently stimulates secretion of cholangiokines that activate liver immune cells and HSCs, thereby promoting liver inflammation and fibrosis.

## Ablation of biliary IKKα does not affect DDC-induced ductular reaction and cholestatic liver injury

To test if IKKα mediates NIK action in cholangiokines, we generated tamoxifen-induced, cholangiocyte-specific IKKα knockout ($IKK\alpha^{\Delta K19}$) mice by crossing $IKK\alpha^{f/f}$ with K19-CreERT drivers. $IKK\alpha^{f/f};K19\text{-}CreERT^{+/-}$ mice were treated with tamoxifen ($IKK\alpha^{\Delta K19}$) or a vehicle (referred to as $IKK\alpha^{f/f}$). Two weeks later, mice were fed a DDC diet for 4 weeks. IKKα was detected in K19⁺ cholangiocytes in $IKK\alpha^{f/f}$ but not $IKK\alpha^{\Delta K19}$ mice by immunostaining (Supplementary Fig. 10A), confirming cholangiocyte-specific IKKα knockout. $IKK\alpha^{\Delta K19}$ mice did not phenocopy $NIK^{\Delta K19}$ mice. Plasma ALT, ALP, and bilirubin levels were indistinguishable between $IKK\alpha^{\Delta K19}$ and $IKK\alpha^{f/f}$ mice (Supplementary Fig. 10B). Cholangiocyte expansion, αSMA⁺ HSC number, and liver fibrosis were also comparable between $IKK\alpha^{\Delta K19}$ and $IKK\alpha^{f/f}$ mice (Supplementary Fig. 10C). Unlike biliary NIK, biliary IKKα is not required for DDC-induced ductular reaction and cholestasis. Because $IKK\alpha^{\Delta K19}$ mice harbor the K19-CreERT transgene, these data indicate that tamoxifen activation of CreERT

does not affect DDC-induced ductular reaction. Therefore, $IKK\alpha^{\Delta K19}$ mice also served as an additional control for $NIK^{\Delta K19}$ mice.

Akt has been known to promote cholangiocyte expansion[3], prompting us to test if it is a downstream mediator of the NIK pathway. We stimulated $NIK^{+/+}$ and $NIK^{-/-}$ cholangiocyte cultures with TWEAK and measured Akt phosphorylation by immunoblotting. TWEAK-stimulated Akt phosphorylation/activation to a higher level in $NIK^{+/+}$ than in $NIK^{-/-}$ cells (Supplementary Fig. 5E). Next, we tested if blocking Akt activation attenuates NIK-stimulated cholangiocyte proliferation, using BrdU assays. Cholangiocyte cultures were pretreated with PI 3-kinase inhibitor wortmannin or LY294002 and then stimulated with TWEAK. TWEAK increased proliferation rates to a higher level in $NIK^{+/+}$ cholangiocytes relative to $NIK^{-/-}$ cholangiocytes; importantly, inhibition of Akt, by either wortmannin or LY29402, abrogated TWEAK-stimulated proliferation of $NIK^{+/+}$ cholangiocytes (Fig. 3F). In contrast, wortmannin and LY29402 did not affect $NIK^{-/-}$ cholangiocyte proliferation. Thus, NIK promotes cholangiocyte expansion at least in part by enhancing Akt activation.

## NIK inhibitor treatment ameliorates DDC-induced ductular reaction, liver injury, and fibrosis

NIK-selective inhibitor Compound 33 (C33) has been developed to treat cancer[29]. We evaluated C33's therapeutic potential in liver disease treatment. To validate C33, we cotransfected HEK293 cells with NF-κB2/p100 and NIK plasmids and then treated them with C33. C33 inhibited cleavage of p100 into p52 (stimulated by NIK) in a dose-dependent manner (Fig. 8A). C33 also inhibited the ability of NIK to stimulate NF-κB luciferase reporter activities (Fig. 8B). To examine C33 action in vivo, we placed C57BL/6 J male mice on a DDC diet for 3 weeks and concurrently treated them with C33 (10 mg/kg body weight, twice a week, a vehicle as control). Liver NF-κB2/p52 levels were considerably lower in C33-treated mice than in vehicle-treated mice but were not eliminated by C33 treatments (Supplementary Fig. 11A), suggesting that C33 partially inhibited NIK under the experimental conditions. C33 treatment significantly lowered plasma ALT, ALP, and bilirubin levels (Fig. 8C). The numbers of K19⁺ cholangiocytes, myeloperoxidase (MPO)-expressing neutrophils, and αSMA⁺ HSCs, and liver fibrosis (Sirius red staining) were significantly lower in C33-treated mice relative to vehicle-treated mice (Fig. 8D, E). Cholangiocytes (Ki67⁺K19⁺) and HSC (Ki67⁺αSMA⁺) proliferation rates were significantly lower in C33-treated mice relative to vehicle-treated mice (Supplementary Fig. 11B, C). In contrast, hepatocyte (HNF4α⁺Ki67⁺) and Kupffer cell (Ki67⁺F4/80⁺) proliferation rates were comparable between C33 and vehicle groups. These results indicate that pharmacological inhibition of NIK is able to alleviate ductular reaction, liver inflammation, and fibrosis in mice.

## Discussion

Ductular reaction commonly occurs in chronic liver diseases, including cholestasis, alcoholic liver disease, and nonalcoholic steatohepatitis (NASH). Ductular reaction associates with poor prognosis and is an important risk factor for intrahepatic cholangiocarcinoma[3,6,30]. Biliary insults, in concert with aberrant liver microenvironments, drive cholangiocyte expansion and ductular reaction. In this study, we have identified cholangiocyte-intrinsic NIK as a molecular bridge connecting biliary insults to ductular reaction, liver injury, inflammation, and

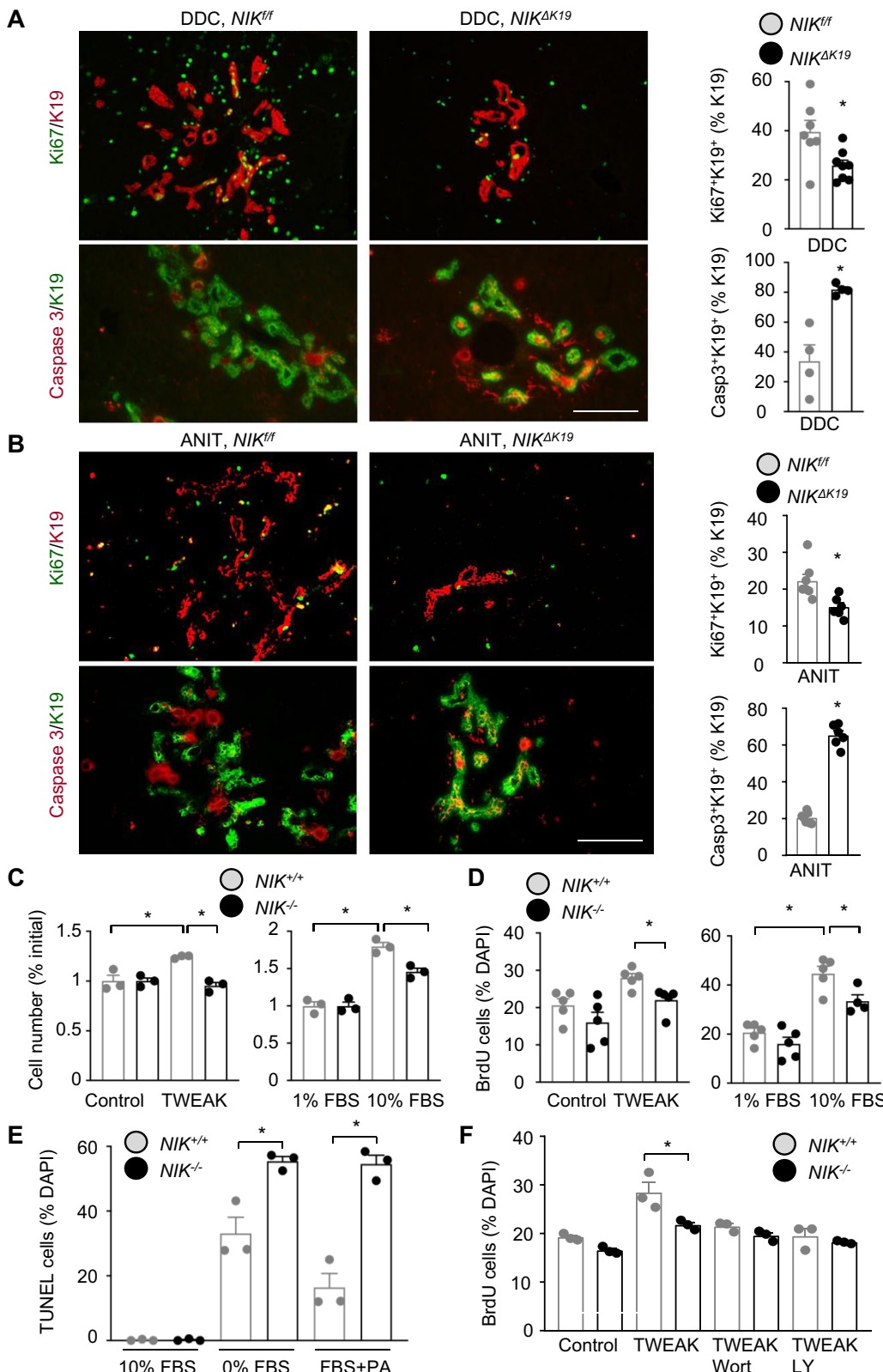

fibrosis. We demonstrated that DDC, ANIT, or BDL treatment markedly upregulated biliary NIK in mice through increasing both its expression and stability. Consistently, DDC and ANIT directly increased the expression and stability of NIK in cholangiocyte cultures. In wild-type mice, DDC, ANIT, or BDL induced ductular reaction, liver injury, inflammation, and fibrosis as expected. Strikingly, cholangiocyte-specific ablation of NIK blocked DDC-, ANIT-, and BDL-induced

ductular reaction, liver injury, and fibrosis in *NIK^ΔK19* mice. Likewise, pharmacological inhibition of NIK by C33 also mitigated DDC-induced ductular reaction, liver injury, inflammation, and fibrosis in mice. Cholangiocyte proliferation was lower while its death was higher in *NIK^ΔK19* mice relative to *NIK^f/f* mice post DDC, ANIT, or BDL treatment, and this may explain the resistance of *NIK^ΔK19* mice to cholangiocyte expansion and ductular reaction. NIK can be activated by cytokines,

**Fig. 3 | NIK directly regulates cholangiocyte proliferation and apoptosis.** $NIK^{\Delta K19}$ and $NIK^{f/f}$ male mice were fed DDC diet for 4 weeks (**A**) or ANIT diet for 3 weeks (**B**). Liver sections were coimmunostained with antibodies to K19 and Ki67 or cleaved caspase-3. Proliferating (Ki67$^+$K19$^+$) and apoptotic (Casp3$^+$K19$^+$) cholangiocytes were counted and normalized to total K19$^+$ cholangiocytes. DDC/Ki67: $NIK^{f/f}$: $n = 7$ mice, $NIK^{\Delta K19}$: $n = 8$ mice; DDC/Casp3: $NIK^{f/f}$: $n = 4$ mice, $NIK^{\Delta K19}$: $n = 4$ mice; ANIT/ Ki67: $NIK^{f/f}$: $n = 7$ mice, $NIK^{\Delta K19}$: $n = 6$ mice; ANIT/Casp3: $NIK^{f/f}$: $n = 7$ mice, $NIK^{\Delta K19}$: $n = 6$ mice. Scale bar: 200 µm. **C** $NIK^{+/+}$ and $NIK^{-/-}$ cholangiocytes were treated for 24 h with FBS or TWEAK (20 ng/ml). Cholangiocytes were counted and normalized to the initial number ($n = 3$ per group). **D** $NIK^{+/+}$ and $NIK^{-/-}$ cholangiocytes were stimulated for 14 h with 10% FBS or TWEAK (10 ng/ml) in the presence of BrdU, followed by immunostaining with anti-BrdU antibody. BrdU$^+$ cholangiocytes were counted and normalized to total cholangiocytes. $NIK^{-/-}$/10% FBS group: $n = 4$ repeats, the other groups: $n = 5$ repeats. **E** $NIK^{+/+}$ and $NIK^{-/-}$ cholangiocytes were deprived of FBS or treated with palmitate (PA) for 24 h, and then subjected to TUNEL assays. TUNEL$^+$ cells were counted and normalized to total cells ($n = 3$ repeats per group). **F** $NIK^{+/+}$ and $NIK^{-/-}$ cholangiocytes were pretreated with wortmannin (Wort) or LY294002 (LY) and then stimulated with TWEAK in the presence of BrdU. BrdU$^+$ cholangiocytes were counted and normalized to total cholangiocytes ($n = 3$ repeats per group). Data are presented as mean ± SEM. *$p < 0.05$, 2-tailed student's $t$-test **A, B, E, F** or 1-way ANOVA **C, D**. Source data are provided as a Source Data file.

intracellular stress, and cell-damaging factors[14,21], raising the possibility that biliary NIK may integrate a wide spectrum of biliary damaging-related signals to promote ductular reaction, liver injury, inflammation, and fibrosis (Fig. 7E). Of note, biliary NIK was profoundly upregulated in humans with PBC, PSC, HBV, HCV, or alcoholic liver disease, raising the intriguing possibility that a similar biliary NIK/ductular reaction/liver inflammation/liver fibrosis cascade may also operate in human liver disease progression.

Given that IKKα acts downstream of NIK to activate the non-canonical NF-κB2 pathway, it is unexpected that $IKK\alpha^{\Delta K19}$ mice did not phenocopy $NIK^{\Delta K19}$ mice. In line with these findings, ablation of cholangiocyte relB, a NF-κB2 partner, also does not affect DDC-induced ductular reaction[31]. Hence, biliary NIK appears to promote ductular reaction by an IKKα/NF-κB2-independent mechanism. We observed that TWEAK stimulated both NF-κB2 and Akt pathways in $NIK^{+/+}$ cholangiocytes. As expected, NIK deficiency completely blocked TWEAK-stimulated activation of the noncanonical NF-κB2 pathway in $NIK^{-/-}$ cholangiocytes. NIK deficiency also substantially attenuated Akt phosphorylation in TWEAK-stimulated $NIK^{-/-}$ cholangiocytes, indicating that TWEAK, or related cytokines, stimulate a NIK/Akt pathway in cholangiocytes. TWEAK has been known to stimulate biliary growth and expansion, and ablation of TWEAK receptor Fn14 suppresses ductular reaction[11]. Akt is also believed to mediate biliary proliferation[3]. Pharmacological inhibition of the PI 3-kinase/Akt pathway blocked TWEAK-stimulated proliferation of $NIK^{+/+}$ cholangiocytes, but not $NIK^{-/-}$ cholangiocytes, suggesting that the biliary NIK/Akt pathway mediates, at least in part, ductular reaction (Fig. 7E).

Ductular reaction has long been recognized to associate with chronic liver disease, but its contribution to liver disease progression remains obscure. Resistance of $NIK^{\Delta K19}$ mice to DDC, ANIT, and BDL-induced liver inflammation and fibrosis strongly argues for the notion that biliary NIK-elicited ductular reaction is a causal factor for liver inflammation and fibrosis. In ductular reaction, reactive cholangiocytes gain proinflammatory and profibrosis properties[3,32]. We observed that expression of inflammatory mediators (Il-1β, Il-4, Il-6, iNos, Tnfα, and Mcp1) and profibrotic genes (Tgfβ1) was markedly lower in $NIK^{-/-}$ cholangiocytes relative to $NIK^{+/+}$ cholangiocytes. In line with these results, macrophages were activated to a higher level by coculture with $NIK^{+/+}$ cholangiocytes relative to $NIK^{-/-}$ cholangiocytes, and to a higher level by stimulation with $NIK^{+/+}$ cholangiocyte-conditioned medium relative to $NIK^{-/-}$ cholangiocyte-conditioned medium. Likewise, $NIK^{+/+}$ cholangiocyte-conditioned medium, which harbored cholangiokines, was much more potent than $NIK^{-/-}$ cholangiocyte-conditioned medium to stimulate HSC activation and its expression of profibrotic genes (αSMA, Col1a1, Mmp9, Ctgf, and Tgfβ1). Based on these findings, we are tempted to postulate that biliary NIK, which is upregulated in chronic liver disease, directly stimulates expression and secretion of cholangiokines that in turn activate liver immune cells and HSCs (Fig. 7E). The biliary NIK/cholangiokine/immune cell axis and the biliary NIK/cholangiokine/HSC axis drive the pathogeneses of liver inflammation and fibrosis. As discussed above, biliary NIK induces ductular reaction, and it is conceivable that expansion of NIK$^+$ cholangiocytes further increases secretion of cholangiokines that shape pathogenic liver microenvironments and exacerbate liver disease progression.

C33 treatment substantially mitigated DDC-induced ductular reaction, liver inflammation, and fibrosis, raising the intriguing possibility that NIK inhibitors are anti-liver disease drug candidates. C33 was not as potent as genetic deletion of biliary *NIK* in protecting against liver disease, because it partially inhibited liver NIK under the experimental conditions. Of note, this study has limitations. We cannot exclude the possibility that C33 may target additional cell types in addition to cholangiocytes in vivo. Global or thymic epithelial cell-specific deletion of *NIK* impairs thymus development and central T cell tolerance in mice[23,24], raising a concern about the safety of NIK inhibitors. However, the human thymus is degenerated in adults, arguing against the possibility that NIK inhibitors may impair T cell-based adaptive immunity in adults. The identities of individual cholangiokines, which mediate crosstalk between cholangiocytes and macrophages and between cholangiocytes and HSCs, remain to be defined. The contribution of the biliary NIK/Akt pathway to ductular reaction needs to be validated in vivo. Human liver biopsies need to be further increased to consolidate the conclusion about biliary NIK in human liver disease. These questions point to future research directions.

## Methods
### Animals
$NIK^{f/f}$, $IKK\alpha^{f/f}$ and *K19-CreERT* mice (C57BL/6J background) were described previously[26,27,33]. *NIK* and *K19* reside in the same Chromosome 11. The $NIK^{f/+}$;$K19$-$CreERT^{-/+}$ genotypes were generated through homologous recombination in germ cell meiosis (low frequency). $NIK^{f/f}$;$K19$-$CreERT$ mice and $IKK\alpha^{f/f}$;$K19$-$CreERT$ mice were treated with tamoxifen (5 mg/mouse, twice, 4 days apart, i.p.) at 7–8 weeks of age to generate $NIK^{\Delta K19}$ and $IKK\alpha^{\Delta K19}$ mice, respectively. Olive oil vehicle was used as control. $NIK^{f/f}$ mice were injected with tamoxifen as an additional control. Mice were housed on a 12-h light-dark cycle and at 25 $^0$C of ambient temperature in the Unit for Laboratory Animal Medicine at the University of Michigan (ULAM) and fed *ad libitum* a chow diet (9% fat in calories; TestDiet, St. Louis, MO), DDC diet, or ANIT diet.

### Ethics statements
Animal research was complied with all relevant ethnic regulations. Animal experiments were conducted following the protocols approved by the University of Michigan Institutional Animal Care and Use Committee (IACUC).

### Human liver biopsies
De-identified human liver specimens were provided by Dr. Suthat Liangpunsakul under the IRB-approved protocol at the Indiana-Purdue University Indianapolis and by Dr. Anna S. Lok under the IRB-approved protocol at the University of Michigan. The study design and conduct were authorized by the Indiana-Purdue University Indianapolis and University of Michigan IRB committees and complied with all relevant regulations regarding the use of human study participants. Patients provided informed consent. The study was conducted in accordance with the criteria set by the Declaration of Helsinki.

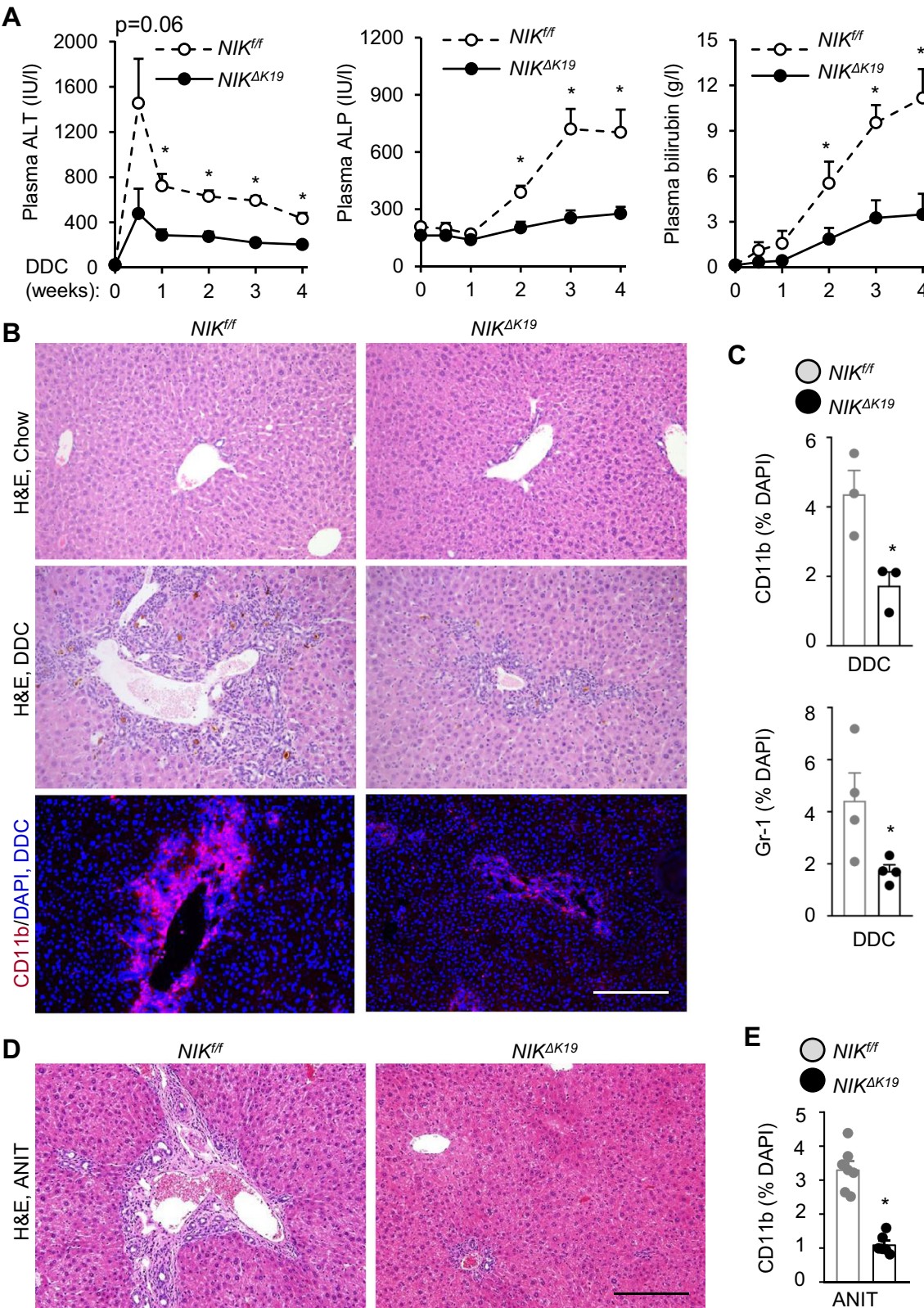

**Fig. 4 | Ablation of biliary NIK ameliorates DDC-induced liver injury and inflammation.** **A**–**C** *NIK^{ΔK19}* and *NIK^{f/f}* male mice were fed a DDC diet for 4 weeks. **A** Plasma ALT, ALP and total bilirubin levels. *NIK^{f/f}*: *n* = 8 mice, *NIK^{ΔK19}*: *n* = 9 mice. **B**, **C** Liver sections were stained with H&E or antibodies to CD11b, and Gr-1. **B** Representative images. Scale bar: 200 μm. **C** CD11b (*n* = 3 mice per group) and Gr-1 cells (*n* = 3 mice per group) were counted and normalized to total liver cell number (*n* = 3–4 mice per group). **D**, **E** *NIK^{ΔK19}* and *NIK^{f/f}* male mice were fed an ANIT diet for 3 weeks. **D** Representative H&E staining of liver sections. Scale bar: 200 μm. **E** Liver sections were stained with anti-CD11b antibody. CD11b cells were counted and normalized to total liver cells. *NIK^{f/f}*: *n* = 7 mice, *NIK^{ΔK19}*: *n* = 6 mice. Data are presented as mean ± SEM. *$p < 0.05$, 2-way ANOVA (**A**) and 2-tailed student's *t*-test (**C**, **E**). Source data are provided as a Source Data file.

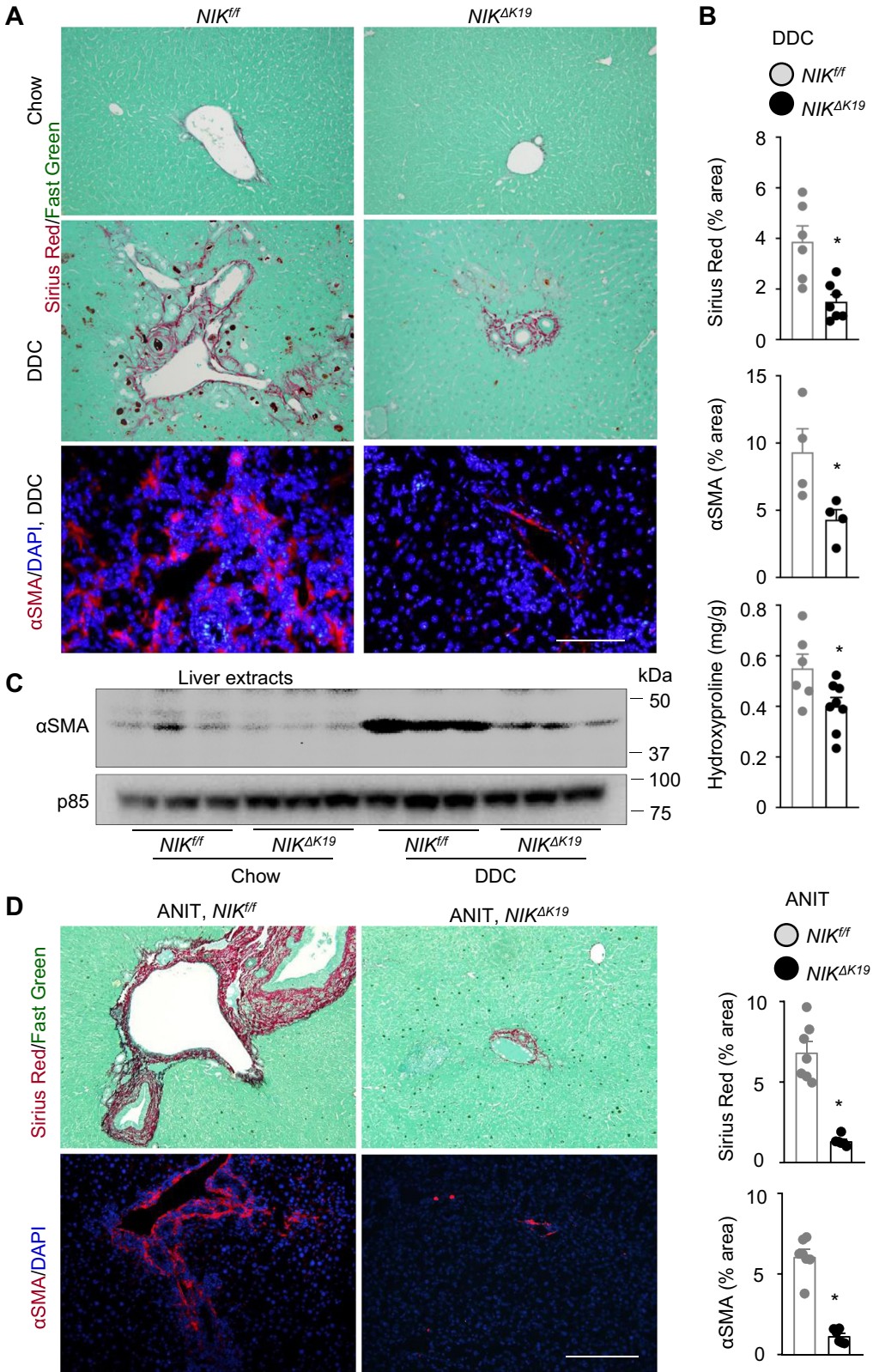

**Fig. 5 | Ablation of NIK in cholangiocytes protects against liver fibrosis.**
**A, B** *NIK*$^{ΔK19}$ and *NIK*$^{f/f}$ male mice were fed a DDC diet for 4 weeks. Liver sections were stained with Sirius red/fast green or antibody to αSMA. **A** Representative images. Scale bar: 200 μm. **B** Sirius red (*NIK*$^{f/f}$: $n = 6$ mice, *NIK*$^{ΔK19}$: $n = 7$ mice) and αSMA HSC ($n = 4$ mice per group) areas were measured and normalized to total areas. Liver hydroxyproline levels were measured and normalized to liver weight (*NIK*$^{f/f}$: $n = 6$ mice, *NIK*$^{ΔK19}$: $n = 8$ mice). **C** Mice were fed a chow or DDC diet for 4 weeks. Liver

extracts were immunoblotted with anti-αSMA and p85 (loading control) antibodies. Each lane represents one individual animal ($n = 3$ mice per group). **D** *NIK*$^{ΔK19}$ ($n = 6$ mice) and *NIK*$^{f/f}$ ($n = 7$ mice) male mice were fed an ANIT diet for 3 weeks. Liver sections were stained with Sirius red/fast green or anti-αSMA. Scale bar: 200 μm. Sirius red and αSMA areas were normalized to total areas. Data are presented as mean ± SEM. *$p < 0.05$, 2-tailed student's *t*-test. Source data are provided as a Source Data file.

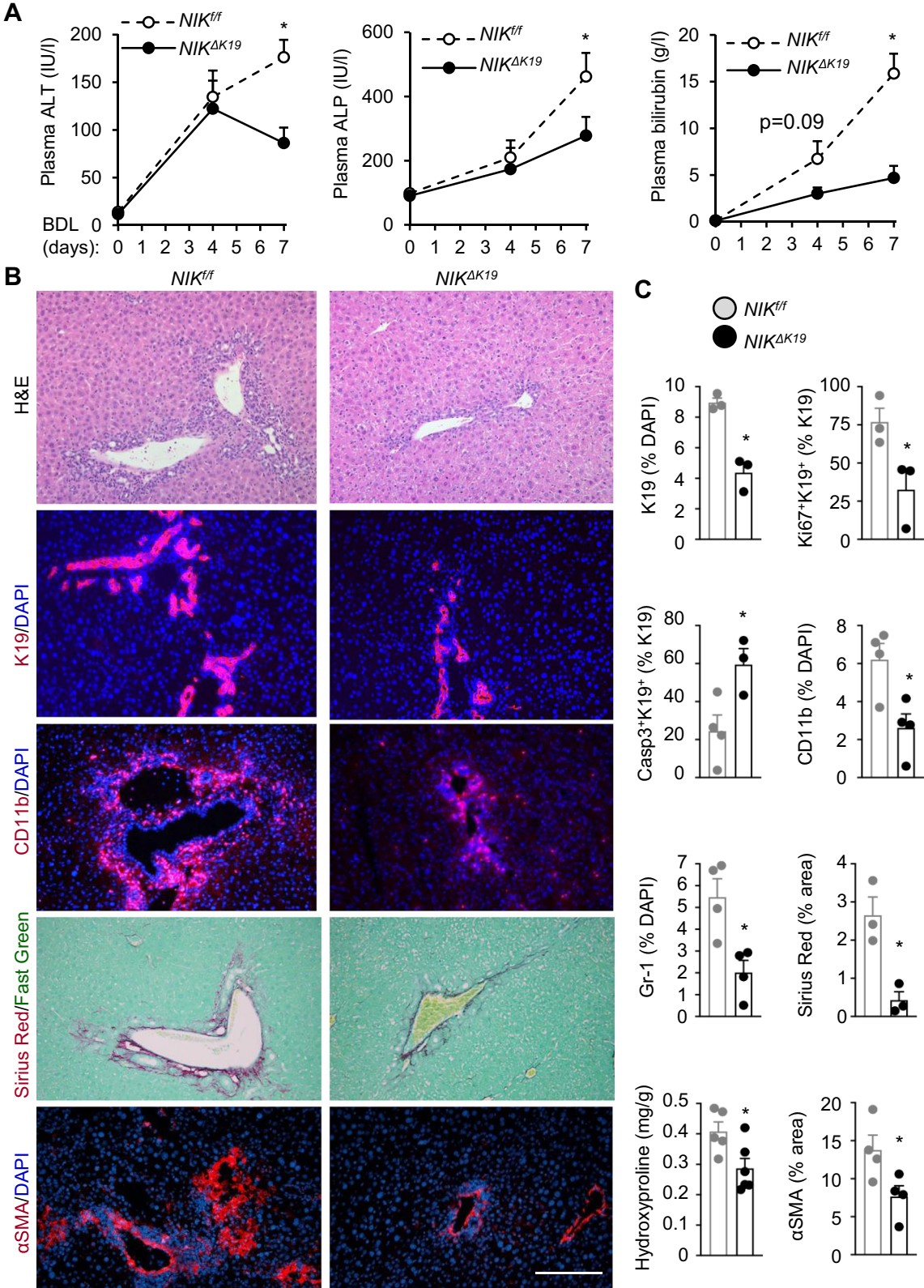

**Fig. 6 | Ablation of biliary NIK attenuates BDL-induced ductular reaction, liver injury, and inflammation.** BDL was performed on $NIK^{\Delta K19}$ and $NIK^{f/f}$ male mice, and livers were analyzed 7 days later. **A** Plasma ALT, ALP, and total bilirubin levels ($n = 6$ mice per group). **B**, **C** Liver sections were stained with the indicated antibodies. **B** Representative images. Scale bar: 200 μm. **C** Individual subpopulations were counted and normalized to total liver cells. Sirius red area was normalized to total

view areas ($n = 3$ mice per group). Liver hydroxyproline content was normalized to liver weight ($NIK^{f/f}$: $n = 5$ mice, $NIK^{\Delta K19}$: $n = 6$ mice). K19: $n = 3$ mice per group; Casp3[+]K19[+]: $n = 3$ mice per group; Ki67[+]K19[+]: $n = 3$ mice per group; Gr-1: $n = 4$ mice per group; CD11b: $n = 4$ mice per group; αSMA: $n = 4$ mice per group. Data are presented as mean ± SEM. *$p < 0.05$, 2-tailed student's $t$-test (**C**) and 2-way ANOVA (**A**). Source data are provided as a Source Data file.

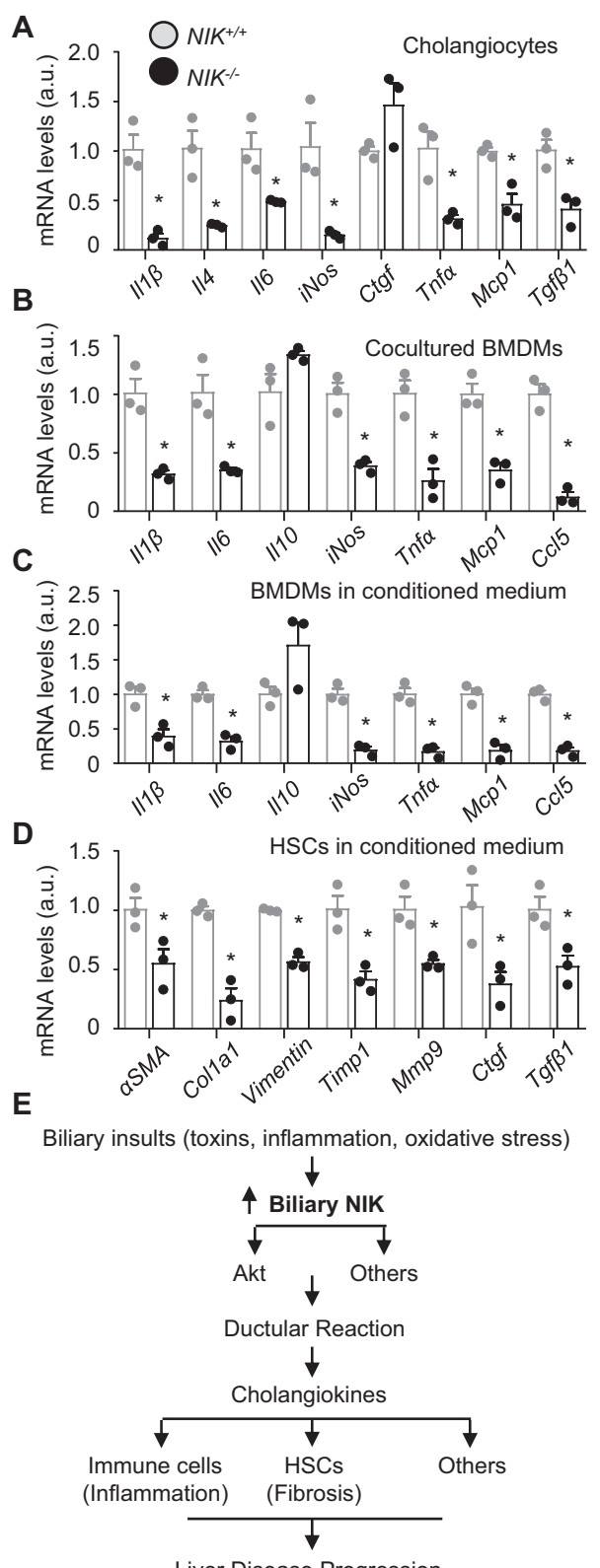

**Fig. 7 | Biliary NIK regulates cholangiocyte-macrophage and cholangiocyte-HSC crosstalk via cholangiokines. A** Cholangiocyte cultures were treated with TWEAK (50 ng/ml) for 48 h. Gene expression was measured by qPCR and normalized to 36B4 expression ($n = 3$ repeats per group). **B** BMDMs were cocultured with TWEAK-stimulated cholangiocytes for 24 h, and BMDM gene expression was measured by qPCR and normalized to 36B4 expression ($n = 3$ repeats per group). **C** BMDMs or **D** HSCs were treated with cholangiocyte-conditioned medium for 6 h and 24 h, respectively. Gene expression was measured by qPCR and normalized to 36B4 levels ($n = 3$ repeats per group). **E** Biliary insults (biliary toxins, oxidative stress, cytokines) activate biliary NIK. NIK induces ductular reaction by enhancing, perhaps via Akt pathways, cholangiocyte proliferation while suppressing cholangiocyte death. Additionally, NIK stimulates secretion of cholangiokines from reactive cholangiocytes. The cholangiokines activate liver immune cells and HSCs, promoting liver inflammation, liver fibrosis, and liver disease progression. Data are presented as mean ± SEM. \**p < 0.05*, 2-tailed student's *t*-test.

### BDL

$NIK^{\Delta K19}$ and $NIK^{f/f}$ mice (9–10 weeks) were anesthetized with isoflurane and subjected to midline laparotomy (-1 cm) to expose the common bile duct. Two knots were made on the common bile duct using a 5–0 silk suture. The bile duct was cut between the two knots. The peritoneum was re-aligned. The skin and the underneath muscle layers were closed individually using a 5–0 silk suture.

### Compound 33 treatment

C57BL/6J male mice (8 weeks) were fed a DDC diet and simultaneously treated with NIK inhibitor Compound 33 (Shanghai Institute of Materia Medica, Chinese Academy of Sciences, Shanghai) at 10 mg/kg body weight (twice a week, i.p.) for 3 weeks. Compound 33 was prepared in a vehicle of 10% PEG400 (polyethylene glycol, molecular weight 400, Sigma #P3265) and 3% Cremophor EL (Sigma #C5135). Body weight, blood chemistry, and liver pathology were analyzed.

### Blood chemistry

Blood samples were collected from tail veins. Plasma ALT, ALP, and bilirubin levels were measured using commercial kits (Pointe Scientific Inc., Canton, MI; ALT: A7526625; ALP: A7516150; bilirubin: B7576120).

### Hydroxyproline assays

Liver samples were homogenized in 6 N HCl, hydrolyzed at 100 °C for 18 h, and centrifuged at 9391xg for 5 min. The supernatant was dried in speed-vacuum, dissolved in $H_2O$, and neutralized with 10 N NaOH. Samples were incubated in a chloramine-T solution [60 mM chloramines-T (Sigma, 857319), 20 mM citrate, 50 mM acetate, pH 6.5] for 25 min at room temperatures, and then in Ehrlich's solution (Sigma, 038910) at 65 °C for additional 20 min. Hydroxyproline content was measured using a Beckman Coulter AD 340 Plate Reader (570 nm) and normalized to liver weight.

### Staining of liver sections

Liver paraffin sections were stained with 0.1% Sirius-red (Sigma, 365548) and 0.1% Fast-green (Sigma, F7252) (in saturated picric acid). Liver frozen sections were prepared using a Leica cryostat (Leica Biosystems Nussloch GmbH, Nussloch, Germany), fixed in 4% paraformaldehyde for 30 min, blocked for 3 h with 5% normal goat serum (Life Technologies) supplemented with 1% BSA, and incubated with the indicated antibodies overnight at 4 °C. Antibodies were listed in Supplementary Table 1. Liver frozen sections were stained with TUNEL reagents using an In Situ Cell Death Detection Kit (Roche Diagnostics, Indianapolis, IN, 11684817910).

### Immunoblotting

Cells or tissues were homogenized in a L-RIPA lysis buffer (50 mM Tris, pH 7.5, 1% Nonidet P-40, 150 mM NaCl, 2 mM EGTA, 1 mM $Na_3VO_4$,

### DDC diet and ANIT diet

Two weeks after tamoxifen treatment (deleting biliary *NIK* or *IKKα*), $NIK^{\Delta K19}$, $IKK\alpha^{\Delta K19}$, and control mice were fed a DDC diet for 4 weeks or with an ANIT diet for 3 weeks. DDC diet and ANIT diet were prepared by adding 0.1% DDC (Sigma-Aldrich, 137030) or 0.08% ANIT (Sigma-Aldrich, 18951) to chow diet powder (LabDiet 5001).

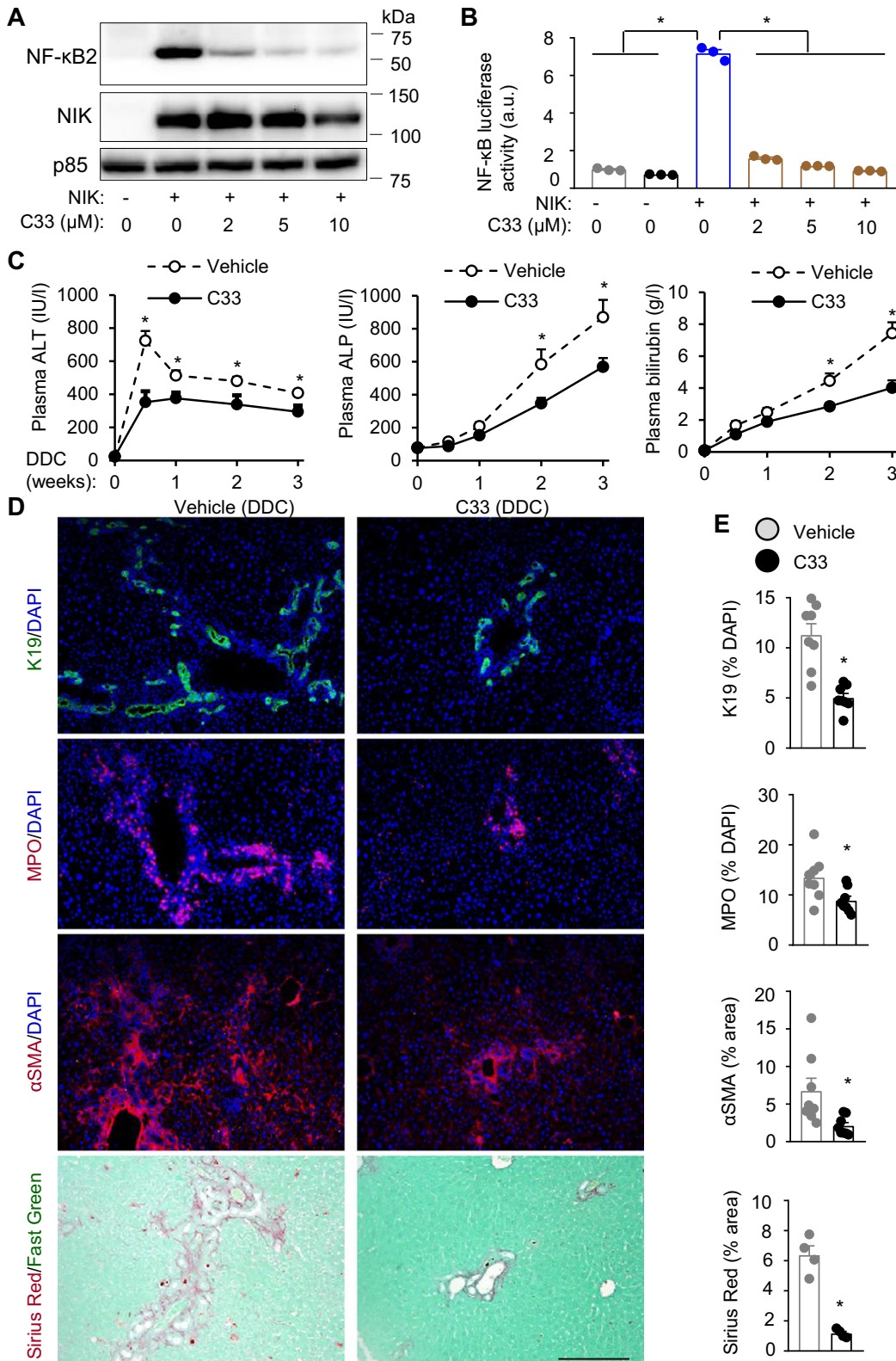

**Fig. 8 | NIK inhibitor treatment ameliorates DDC-induced ductular reaction and liver injury. A** HEK293 cells were cotransfected with NF-κB2 and NIK expression plasmids and treated with C33. Cell extracts were immunoblotted with antibodies to NF-κB2 and NIK. **B** HEK293 cells were cotransfected with NF-κB luciferase reporter and NIK expression plasmids and treated with C33. Luciferase activity was assessed 48 h after C33 treatment (*n* = 3 repeats per group). **C–E** C57BL/6 J males were treated with DDC and C33 for 3 weeks. **C** Plasma ALT, ALP, and total bilirubin levels (*n* = 8 mice per group). **D**, **E** Liver sections were immunostained with the indicated antibodies. **D** Representative liver images. Scale bar: 200 µm. **E** K19⁺ and MPO⁺ cells were counted and normalized to total liver cells. Sirius red and αSMA HSC areas were normalized to total areas. K19, MPO, and αSMA: *n* = 8 mice per group, Sirius red: *n* = 4 mice per group. Data are presented as mean ± SEM. *$p < 0.05$, 2-tailed student's *t*-test (**E**) and 1-way (**B**) or 2-way (**C**) ANOVA. Source data are provided as a Source Data file.

100 mM NaF, 10 mM $Na_4P_2O_7$, 1 mM benzamidine, 10 µg ml$^{-1}$ aprotinin, 10 µg ml$^{-1}$ leupeptin, 1 mM phenylmethylsulfonyl fluoride). Proteins were separated by SDS-PAGE and immunoblotted with the indicated antibodies. Antibodies were listed in Supplementary Table 1. HEK293 cell lines were from the ATCC (CRL-1573).

### Isolation of the biliary tree and generation of cholangiocyte lines

*NIK$^{f/f}$* males (9 weeks) were anesthetized with isoflurane, and the liver was perfused with type II collagenase (Worthington Biochem, Lakewood, NJ). Bile duct trees were isolated, extensively washed with PBS, weighed, minced into ~1 mm blocks, and cultured in DMEM (25 mM glucose) supplemented with 10% FBS and antibiotics. Proliferating cholangiocytes, portal fibroblasts, and HSCs emigrated from tissue blocks in 7–8 days. These cells were collected, transduced with E1A lentiviral vectors, and treated with puromycin for 1 week to select immortalized cells. Individual lines, derived from single cell lineage, were generated by a series of dilution. To characterize these lines, we measured expression of expression of cholangiocyte markers (K7, K19, Cftr, Hnf1β) and HSC markers (col-1a1, αSMA) by qPCR. *NIK$^{f/f}$* cholangiocytes (*NIK$^{+/+}$*) were transduced with Cre adenoviral vectors to delete *NIK* (*NIK$^{-/-}$*). Adenoviral vectors were subsequently removed by repeatedly passaging (>10 generations). Cholangiocytes were cultured in DMEM supplemented 10% FBS and antibiotics.

### Cholangiocyte proliferation and death

Cholangiocyte cultures (~50% confluence) were deprived of serum overnight and stimulated with 1% FBS, 10% FBS, or TWEAK (20 ng/ml) for 24 h. Cells were counted. Cholangiocytes were stimulated for 24 h with FBS or TWEAK in the presence of BrdU and stained with anti-BrdU antibody. Cholangiocytes (~80% confluence) were cultured for 24 h in DMEM without FBS or in DMEM supplemented with 10% FBS and 200 µM palmitate. Cholangiocyte death and survival were assessed by TUNEL or MTT assays.

### Cholangiocyte-conditioned medium and cholangiocyte/BMDM cocultures

Confluent *NIK$^{+/+}$* and *NIK$^{-/-}$* cholangiocytes were cultured for 8 h in DMEM supplemented with 1% FBS and 50 ng/ml TWEAK. Culture medium was replaced with fresh DMEM (1% FBS, no TWEAK) and collected 24 h later (cholangiocyte-conditioned medium). BMDMs were isolated from the femur and tibia of C57BL/6J male mice and cultured in DMEM supplemented with 20% L929 conditioned medium, 10% heat-inactivated FBS, 100 units/ml penicillin, and 100 µg/ml streptomycin[15]. BMDMs were fully differentiated 6 days later. Differentiated BMDMs were grown in cholangiocyte-conditioned medium for 6 h, and their gene expression was assessed by qPCR. HSCs were cultured in DMEM supplemented with 5% FBS, 100 units/ml penicillin, and 100 µg/ml streptomycin. Confluent HSCs were grown for 24 h in cholangiocyte-conditioned medium supplemented with 50 ng/ml TGFβ1, and their gene expression was measured by qPCR. In transwells, cholangiocytes were seeded in the insert and pretreated with DMEM supplemented with 1% FBS and 50 ng/ml TWEAK for 8 h and then cocultured with BMDMs grown in DMEM supplemented 20% L929 conditioned medium, 10% heat-inactivated FBS, 100 units/ml penicillin, and 100 µg/ml streptomycin in the bottom well. BMDMs were harvested 24 h after coculture with cholangiocytes, and their gene expressions were measured by qPCR.

### NIK stability in cholangiocyte cultures

Cholangiocytes were transduced with β-galactosidase (β-gal) or NIK adenoviral vectors for 2 h and subsequently grown for 16 h in DMEM supplemented with 5% FBS. The medium was replaced with DMEM supplemented with DDC (100 uM), ANIT (50 uM) or DMSO, and 4 h later, cells were treated with cycloheximide (100 ug/ml) for 0.5, 1, 1.5, 2, 2.5 and 3 h. Cell extracts were prepared and immunoblotted with antibodies to NIK and β-actin. NIK levels were quantified, normalized to β-actin, and presented as a percentage of the initial values.

### Real-time quantitative PCR (qPCR)

Total RNAs were extracted using TRIzol reagents (Life technologies). Relative mRNA abundance was measured using Radiant™ SYBR Green 2X Lo-ROX qPCR Kits (Alkali Scientific, Pompano Beach, FL) and StepOnePlus RT PCR Systems (Life Technologies Corporation, NY, USA). Primers were listed in Supplementary Table 2.

### Statistics

Data were analyzed using Microsoft excel 2016 and presented as means ± sem. The difference was analyzed between two groups by two-tailed Student's *t*-test and between multiple groups by 1-way ANOVA/Tukey test (GraphPad Prism 8). Longitudinal data were analyzed by 2-way ANOVA/Sidak posttest (GraphPad Prism 8). $P < 0.05$ was considered statistically significant.

### Reporting summary

Further information on research design is available in the Nature Research Reporting Summary linked to this article.

## Data availability

Source data are provided within this paper as a Source Data file. All data supporting the findings described in this paper are available in the article and in the Supplementary Information and from the corresponding author upon reasonable request. Source data are provided with this paper.

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

## Acknowledgements

We thank Drs. Yatrik Shah, Xin Tong, and Lei Yin (University of Michigan) for helpful discussions. This study was supported by grants R01 DK114220, R01 DK115646, R01 DK127568, R01 DK130111, and R21 AA025945 (L.R.), R01 DK116548 (M.B.O.), UH3 AA026903 (S.L.) from the National Institutes of Health, and University of Michigan GI Innovation Fund. This work utilized the cores supported by the Michigan Diabetes Research and Training Center (NIH DK020572), Michigan Metabolomics and Obesity Center (DK089503), and the University of Michigan Gut Peptide Research Center (NIH DK34933).

## Author contributions

Z.Z., X.Z., H.S., and L.S. conducted the experiments, X.Z., H.S., and L.R. designed the experiments and wrote the paper, X.Z., H.S., Z.Z., L.S., S.L., A.S.L., M.B.O., S.W., and L.R. edited the paper, and S.L., A.S.L., and M.B.O. provided human liver biopsies.

## Competing interests
The authors declare no competing interests.
