## [Peer Review File · Nature Communications]

Reviewers' Comments:

Reviewer #1:

Remarks to the Author:

Zhong and colleagues utilize biliary specific knock of NF-KappaB inducing kinase (NIK) to demonstrate the role of NIK in the promotion of ductular reaction, liver injury, inflammation and fibrosis in two different models of Cholestasis (DDC diet and BDL) in mice. The study provides extensive data in the two models that show a reduction in ductular reaction and other parameters. That being said the study is somewhat descriptive in design and do not clearly demonstrate a mechanism by which NIK regulates biliary proliferation.

1. It is not clear in the text the rational for utilizing palmitate in the in vitro studies.
2. The authors have previously shown that NIK suppressed hepatocyte regeneration. The authors need to address the potential role of NIK remaining in hepatocytes and potentially other cell types that may contribute to the alterations in cholangiocyte proliferation. Hepatocyte response to DDC may be altered due to lack of NIK in cholangiocytes and vice versa. Are the levels of NIK altered in hepatocytes and other cell types in the cholangiocyte specific knockout model during cholestasis?
3. In the NIK agonist treated mice, were there differences in hepatocyte responses (i.e., proliferation) since NIK is expressed in hepatocytes?

Reviewer #2:

Remarks to the Author:

In submitted manuscript, the authors present the data which suggest the promotion of bile ductal abnormalities by NIK. However, the molecular mechanism underlying NIK activity in this case is not clear as this NIK activity is not mediated through non-canonical NF-kB signaling.

Therefore, the authors need to provide these critical mechanistic data for NIK activity: which signaling pathway(s) is/are regulated by biliary NIK in cholangiocytes. They should also address a number of deficiency by providing additional data as stated below.

1. The mode of regulation of NIK expression in different pathological conditions shown in Fig.1A should be examined further.
 - What are the levels of mRNA expression as compared to control samples;
 - Are NIK levels regulated on posttranslational level? In this regard, the specificity of NIK staining shown in figure 1 should be confirmed. NIK expression is regulated tightly by the complex of the proteins containing E3 ligases cIAP1 and cIAP2. cIAP1 and 2 constitutively ubiquitinate NIK propelling its proteasomal degradation. Typically, signal transduction pathways triggered by some TNF family members such as LIGHT, TWEAK or BAFF trigger dissociation and/or proteasomal degradation of cIAP-containing complex from NIK enabling its stabilization, accumulation and activation. Is it the case here? The authors should examine by western blotting the expression of TRAF2, TRAF3, cIAP1, cIAP2, p100/p52 and NIK in samples presented in figure 1A and C.
 - Importantly, specificity of NIK antibody should be evaluated in cells and tissues lacking NIK expression.

Unfortunately, the production of NIK-specific antibodies used by the authors is discontinued. Therefore, the data presented in the Fig.1 cannot be independently reproduced. Authors should validate NIK expression in mice samples presented in Fig.1C using alternative antibodies with validated specificity.

2. The nature of the top protein shown in Fig.2A is not clear. Is it p52? Please show clearly p100 and p52 levels and provide molecular weight markers. Moreover, please provide the NIK staining of DDC and control samples (as it was shown in Fig. 1C).
3. The authors presented evidence for NIK-dependent but IKK1/a-independent development of ductal reaction. Additionally, it was previously shown that this is independent of NF-kB2 (citation 28 in the current manuscript). In this case, the relevance of the data showing cholangiocytes proliferation (Fig. 3) in response to TWEAK that signal through activation of non-canonical NF-kB pathway is not clear. And how do FBS levels mediate NIK activation?
4. The amelioration of liver enzymes achieved by DCC mice treatment with NIK inhibitor C33 shown in Fig. 7E is significantly lower as compared with data obtained with NIK-null mice (Fig.

4A). Is it because C33 affects mostly non-canonical NF- κ B pathway but not other pathways regulated by NIK? This argues against safe and efficacious use of NIK inhibitor C33 (or other NIK inhibitors) in liver diseases given the pleiotropic nature of NIK function/activity. Finally, why and how do increased doses of C33 affect NIK levels in figure 7A?

Reviewer #3:

Remarks to the Author:

The manuscript by Zong et al investigates the role of NF κ B-inducing kinase (NIK) in the ductular reaction and seeks to determine whether NIK inhibition could be an effective treatment in diseases where the ductular reaction is excessive.

There are a number of typos in the manuscript and at points it is difficult to read - as a result the manuscript ends up feeling quite list-like in its presentation and would benefit from a rewrite to improve the English and the accessibility of the manuscript.

Regarding the data, there are some interesting observations presented here, however I found the manuscript to be quite preliminary and some of the data should be explored further to properly understand how NIK is driving the ductular reaction and BEC proliferation.

Specific points:

1. The N number in the human experiment is very small and I would struggle to be convinced by the conclusions drawn from this data (particularly given the variability in human pathology) Whilst the authors state that there is an N=3 per group, in figure 1B, the control group and the HBV group appear to have N=2 per group - it is not clear to me how the authors managed to generate significance values using the statistical tests identified in their materials and methods. have the authors tested for normality and how have they come to use the tests described?

2. BDL is not a great model of cholestasis and whilst it does give you some BEC proliferation and inflammation it is very artificial and the hyperproliferation seen here does not reflect any human pathologies - the authors should consider using a different portal injury model such as Thioacetamide or Furin to model the ductular reaction.

3. In figure 1A and 1C/E, the expression of NIK is not localised to the BECs alone, rather there is extensive NIK staining in other cell types. Given this manuscript goes on to demonstrate that C33 inhibits the formation of a stromal microenvironment around the BECs, you should characterise which other cells are expressing NIK.

4. I am concerned by the controls used in the K19NIK-delta model. Rather than using animals which have the CreERT and a WT NIK allele you have chosen to use an uninduced (un treated with tamoxifen) group of animals which would not account for non-specific activity of the Cre recombinase in this system. nor would it account for the fact that tamoxifen alone is known to alter the proliferation of BECs - the latter is more important perhaps given that you are assaying BEC proliferation and we know from other CreERT systems that the half-life of Tamoxifen can be protracted. It is also not clear from the methods whether animals receive DDC directly following tamoxifen treatment or whether they are rested between tamoxifen and DDC treatment.

5. The authors use a very poor WB to measure the levels of NF- κ B2, this needs to be improved if it is to be used as its very variable (as is the p85 loading control). Also, what is the evidence that NF κ B2 is a robust readout of NIK activity as from the literature it could be regulated by a number of signals?

6. The data in Fig 2B is confusing - there is no description of how you isolate the biliary tree in the methods, nor do you know that you have extracted the biliary tree completely from all animals - I would suggest that this is a non-standard method and either needs to be more completely validated as a novel method or removed from the manuscript.

7. Throughout the manuscript (particularly in the NIK knockout model) there is a lot of non-epithelial proliferation and apoptosis however you do not account for this? You recognise in the text that the relationship between BECs and the stroma is a complex one and likely interrelated, but fail to address clear changes in the microenvironment when BEC NIK is deleted.

8. You should characterise the cholangiocyte lines generated in SF1B more completely. K19 staining alone is insufficient.

9. It is not clear from the manuscript how TWEAK regulates NFkB2 - you should explain the relationship here more - why not treat with TNF?

10. The authors describe changes in the immune microenvironment in NIK KO BECs. How does this work? what is the cytokine/chemokine profile of the BECs with and without NIK - it was recently reported that TWEAK induced MCP1 expression in cholangiocarcinoma, is this the same mechanism here?

11. The authors do not really investigate how NIK is playing a role in the microenvironment more generally - this should be explored. Is the reduction in fibroblasts/fibrosis/immune cells due to a reduction in epithelial cells or is this independent of epithelial number - if the latter, what is being produced by the WT epithelium that is stimulating the formation of the microenvironment.

12. What is the recombination efficiency of the K19Cre line and have the authors confirmed that in these cells there is always loss of both the IKKa allele and NIK allele at the same frequency?

13. If NIK is not acting through IKKa, how is it acting and what is the mechanism here?

Response to Reviewer #1:

“Zhong and colleagues utilize biliary specific knock of NF-KappaB inducing kinase (NIK) to demonstrate the role of NIK in the promotion of ductular reaction, liver injury, inflammation and fibrosis in two different models of Cholestasis (DDC diet and BDL) in mice. The study provides extensive data in the two models that show a reduction in ductular reaction and other parameters. That being said the study is somewhat descriptive in design and do not clearly demonstrate a mechanism by which NIK regulates biliary proliferation”.

We examined Akt pathways, following these comments. Deletion of *NIK* suppressed Akt phosphorylation in cholangiocytes in response to TWEAT stimulation (new Supplemental Fig. S5E). Pharmacological inhibition of the PI3K/Akt pathway, via wortmannin or LY294002, blocked the ability of TWEAK to stimulate the proliferation of *NIK*^{+/+}, but not *NIK*^{-/-}, cholangiocytes (new Fig. 3F). These new results suggest that the NIK/PI 3-kinase/Akt pathway mediates, at least in part, biliary proliferation. We will validate this pathway in vivo in the future (beyond the scope of this paper). We wish to point out that this paper uncovers, for the first time, the pivotal role of biliary NIK in the ductular reaction, liver injury, inflammation, and fibrosis. It also provides proof of principle evidence that NIK inhibitors have promising translational potential. These findings are expected to be interesting to many investigators studying liver disease.

1. *“It is not clear in the text the rational for utilizing palmitate in the in vitro studies”.*

We provided additional explanation about palmitate (cited new references) following these comments. Liver saturated fatty acids are increased in NASH, alcoholic liver disease, and other types of chronic liver disease. Palmitate has been reported to induce cholangiocyte injury in vitro. It is reasonable to test if NIK deficiency aggravates palmitate toxicity.

2. *“The authors have previously shown that NIK suppressed hepatocyte regeneration. The authors need to address the potential role of NIK remaining in hepatocytes and potentially other cell types that may contribute to the alterations in cholangiocyte proliferation. Hepatocyte response to DDC may be altered due to lack of NIK in cholangiocytes and vice versa. Are the levels of NIK altered in hepatocytes and other cell types in the cholangiocyte specific knockout model during cholestasis”?*

We measured NIK in hepatocytes and immune cells in DDC-fed mice, following these comments. NIK levels in hepatocyte and macrophages were comparable between *NIK*^{ΔK19} and *NIK*^{ΔK19} mice post DDC diet (new Supplemental Fig. S3E-F), arguing against the possibility that hepatocyte and macrophage NIK promote ductular reaction under our experimental conditions. Further supporting the conclusion that cholangiocyte-intrinsic *NIK* promotes biliary expansion, TWEAK or FBS directly stimulated, in the absence of hepatocytes or other cell types, proliferation of *NIK*^{+/+} cholangiocyte cultures to a higher level relative to *NIK*^{-/-} cholangiocyte cultures.

3. *“In the NIK agonist treated mice, were there differences in hepatocyte responses (i.e., proliferation) since NIK is expressed in hepatocytes”?*

We conducted the requested experiments. Hepatocyte proliferation rates were comparable between C33-treated and vehicle-treated mice post DDC diet (new Supplemental Fig. S11B-C). Of note,

NIK levels were lower in hepatocytes relative to cholangiocytes post DDC diet, which may explain the distinct responses of cholangiocytes vs hepatocytes to C33 treatment.

Response to Reviewer #2:

“In submitted manuscript, the authors present the data which suggest the promotion of bile ductal abnormalities by NIK. However, the molecular mechanism underlying NIK activity in this case is not clear as this NIK activity is not mediated through non-canonical NF- κ B signaling. Therefore, the authors need to provide these critical mechanistic data for NIK activity: which signaling pathway(s) is/are regulated by biliary NIK in cholangiocytes. They should also address a number of deficiency by providing additional data as stated below”.

1. *“The mode of regulation of NIK expression in different pathological conditions shown in Fig.1A should be examined further”.*

“- What are the levels of mRNA expression as compared to control samples”

We requested additional human liver samples from our collaborators; unfortunately, our collaborates do not have human liver biopsies for us to conduct qPCR assays. We will continue to look for human liver biopsies and measure NIK mRNA levels in the future.

“- Are NIK levels regulated on posttranslational level? In this regard, the specificity of NIK staining shown in figure 1 should be confirmed. NIK expression is regulated tightly by the complex of the proteins containing E3 ligases cIAP1 and cIAP2. cIAP1 and 2 constitutively ubiquitinate NIK propelling its proteasomal degradation. Typically, signal transduction pathways triggered by some TNF family members such as LIGHT, TWEAK or BAFF trigger dissociation and/or proteasomal degradation of cIAP-containing complex from NIK enabling its stabilization, accumulation and activation. Is it the case here? The authors should examine by western blotting the expression of TRAF2, TRAF3, cIAP1, cIAP2, p100/p52 and NIK in samples presented in figure 1A and C”.

We were unable to obtain human liver biopsies for immunoblotting and qPCR assays. As an alternative approach, we measured expression of these proteins in cholangiocyte cultures and mouse livers, as suggested. DDC stimulation decreased cIAP1 and cIAP2 in cholangiocyte cultures (Supplemental Fig. S2G). Likewise, DDC or ANIT feeding also decreased liver cIAP1 and cIAP2 protein levels in vivo (Supplemental Fig. S2H-I). Traf2 and Traf3 levels are upregulated by DDC, likely due to a compensation to cIAP1/2 deficiency. The anti-NIK antibody was unable to detect endogenous NIK by immunoblotting. The p100/p52 immunoblotting was presented in Fig. 2C.

“- Importantly, specificity of NIK antibody should be evaluated in cells and tissues lacking NIK expression. Unfortunately, the production of NIK-specific antibodies used by the authors is discontinued. Therefore, the data presented in the Fig.1 cannot be independently reproduced. Authors should validate NIK expression in mice samples presented in Fig.1C using alternative antibodies with validated specificity”.

We validated anti-NIK antibody in the livers of whole body NIK knockout mice as suggested. Mice were fed a DDC diet for 2 weeks to increase expression of endogenous NIK. Liver sections were

immunostained with anti-NIK antibody. The anti-NIK antibody detected NIK in *NIK*^{+/+}, but not *NIK*^{-/-}, mice (new Supplemental Fig. S1A).

2. *“The nature of the top protein shown in Fig.2A is not clear. Is it p52? Please show clearly p100 and p52 levels and provide molecular weight markers. Moreover, please provide the NIK staining of DDC and control samples (as it was shown in Fig. 1C)”*

We replaced the original Fig. 2A with the revised Fig. 2C, showing p100, p52, and molecular weight markers as requested. We performed NIK staining of chow and DDC-fed samples as suggested. NIK was detected in K19⁺ cholangiocytes from DDC-fed *NIK*^{fl/fl}, but not *NIK*^{ΔK19}, mice (new Fig. 2A-B).

3. *“The authors presented evidence for NIK-dependent but IKK1/a-independent development of ductal reaction. Additionally, it was previously shown that this is independent of NF-kB2 (citation 28 in the current manuscript). In this case, the relevance of the data showing cholangiocytes proliferation (Fig. 3) in response to TWEAK that signal through activation of non-canonical NF-kB pathway is not clear. And how do FBS levels mediate NIK activation?”*

The noncanonical IKKα/NF-kB pathway was used to further confirm the deletion of *NIK* in *NIK*^{-/-} cholangiocytes. FBS contains diverse cytokines and growth factors that activate NIK.

To gain insight into potential molecular mechanisms, following these comments, we examined the PI 3-kinase/Akt pathway in cholangiocyte cultures. The Akt pathway has been known to promote cholangiocyte expansion. Deletion of *NIK* decreased TWEAK-stimulated Akt phosphorylation (new Supplemental Fig. S5E). Pharmacological inhibition of the PI3K/Akt pathway, via wortmannin or LY294002, blocked the ability of TWEAK to stimulate the proliferation of *NIK*^{+/+}, but not *NIK*^{-/-}, cholangiocytes (new Fig. 3F). Thus, the NIK/PI 3-kinase/Akt pathway promotes biliary proliferation. We will further test the biliary NIK/Akt pathway in mice in the future (beyond the scope of this paper).

4. *“The amelioration of liver enzymes achieved by DCC mice treatment with NIK inhibitor C33 shown in Fig. 7E is significantly lower as compared with data obtained with NIK-null mice (Fig. 4A). Is it because C33 affects mostly non-canonical NF-kB pathway but not other pathways regulated by NIK? This argues against safe and efficacious use of NIK inhibitor C33 (or other NIK inhibitors) in liver diseases given the pleiotropic nature of NIK function/activity. Finally, why and how do increased doses of C33 affect NIK levels in figure 7A?”*

The synthesis and purification of C33 are time-consuming and expensive, and we have produced a limited amount of C33 for the experiments. We do not have enough C33 to perform dose response assays. We assessed C33 inhibition of liver NIK, using p52 production as a surrogate marker. C33 partially inhibited NIK in C33-treated mice under our experimental conditions (new Supplemental Fig. S11A). The reduced responses of C33-treated mice can be explained by partial inhibition of biliary NIK. The C33 results in Fig. 7 and new Supplemental Fig. S11 are expected to stimulate a strong interest to further test the safety, efficacy, and translational significance of NIK inhibitors in the future.

We repeated the experiments in Fig. 7A several times and found that C33 did not affect NIK levels at 2-5 uM. At 10 uM, C33 appeared to decrease NIK levels, and the underlying mechanism

remains unclear. Perhaps C33 at an extremely high dose induces NIK degradation, which will be tested in the future.

Response to Reviewer #3:

“The manuscript by Zong et al investigates the role of NFκB-inducing kinase (NIK) in the ductular reaction and seeks to determine whether NIK inhibition could be an effective treatment in diseases where the ductular reaction is excessive. There are a number of typos in the manuscript and at points it is difficult to read - as a result the manuscript ends up feeling quite list-like in its presentation and would benefit from a rewrite to improve the English and the accessibility of the manuscript. Regarding the data, there are some interesting observations presented here, however I found the manuscript to be quite preliminary and some of the data should be explored further to properly understand how NIK is driving the ductular reaction and BEC proliferation”.

Specific points:

1. *“The N number in the human experiment is very small and I would struggle to be convinced by the conclusions drawn from this data (particularly given the variability in human pathology) Whilst the authors state that there is an N=3 per group, in figure 1B, the control group and the HBV group appear to have N=2 per group - it is not clear to me how the authors managed to generate significance values using the statistical tests identified in their materials and methods. have the authors tested for normality and how have they come to use the tests described”?*

It has been extremely difficult for us to obtain human liver biopsies. Nonetheless, we obtained additional one control (n=3) and three HBV samples (n=5). We added the new results to the statistical analyses.

2. *“BDL is not a great model of cholestasis and whilst it does give you some BEC proliferation and inflammation it is very artificial and the hyperproliferation seen here does not reflect any human pathologies - the authors should consider using a different portal injury model such as Thioacetamide or Furin to model the ductular reaction”.*

We added a new α-naphthyl-isothiocyanate (ANIT) models (new Fig. 2G, Fig. 3B, Fig. 4D-E, Fig. 5D, Supplemental Fig. S3G, and Supplemental Fig. S7A-B), in addition to BDL and DDC models, following these comments. We did not use thioacetamine or furin because they may target non-biliary cells, complicating data interpretation. ANIT has been reported to specifically target cholangiocytes (references cited). *NIK*^{ΔK19} mice were resistant to ANIT-, DDC-, and BDL-induced ductular reaction, liver inflammation, and fibrosis.

3. *“In figure 1A and 1C/E, the expression of NIK is not localised to the BECs alone, rather there is extensive NIK staining in other cell types. Given this manuscript goes on to demonstrate that C33 inhibits the formation of a stromal microenvironment around the BECs, you should characterise which other cells are expressing NIK”.*

We did NIK co-staining of liver sections with K19, HNF4 α (hepatocytes), and CD11b (macrophages), following these comments. HNF4 α ⁺NIK⁺ and CD11b⁺NIK⁺ cell frequencies were comparable between *NIK*^{AK19} and *NIK*^{fl/fl} mice post DDC diet (Supplemental Fig. S3E-F).

4. *“I am concerned by the controls used in the K19NIK-delta model. Rather than using animals which have the CreERT and a WT NIK allele you have chosen to use an uninduced (un treated with tamoxifen) group of animals which would not account for non-specific activity of the Cre recombinase in this system. nor would it account for the fact that tamoxifen alone is known to alter the proliferation of BECs - the latter is more important perhaps given that you are assaying BEC proliferation and we know from other CreERT systems that the half-life of Tamoxifen can be protracted. It is also not clear from the methods whether animals receive DDC directly following tamoxifen treatment or whether they are rested between tamoxifen and DDC treatment”.*

We clarified confusions about the controls. Three controls were used for *NIK*^{AK19} mice. 1) *NIK*^{fl/fl};K19-CreERT mice were treated with a vehicle. 2) *NIK*^{fl/fl} mice were treated with tamoxifen (Supplemental Fig. S3B). Tamoxifen treatment alone did not influence the effect of DDC on the liver. 3) *IKK α* ^{fl/fl};K19-CreERT mice were treated with tamoxifen (new Supplemental Fig. S10A-C). Tamoxifen activation of CreERT mice did not influence the effect of DDC on the liver.

We clarified in the method that 2 weeks after tamoxifen treatment, the mice were fed a DDC diet.

5. *“The authors use a very poor WB to measure the levels of NF- κ B2, this needs to be improved if it is to be used as its very variable (as is the p85 loading control). Also, what is the evidence that NF κ B2 is a robust readout of NIK activity as from the literature it could be regulated by a number of signals”?*

We replaced the original NF- κ B2 immunoblotting and used β -actin (not p85) as a loading control (new Fig. 2C). Following these comments, we costained liver sections with antibodies to NIK and K19, confirming that cholangiocyte NIK is deleted in *NIK*^{AK19} mice compared to *NIK*^{fl/fl} mice (new Fig. 2A-B).

6. *“The data in Fig 2B is confusing - there is no description of how you isolate the biliary tree in the methods, nor do you know that you have extracted the biliary tree completely from all animals - I would suggest that this is a non-standard method and either needs to be more completely validated as a novel method or removed from the manuscript”.*

We added the requested methods and moved the original Fig. 2B to the new Supplemental Fig. S3D, following these comments. The biliary tree was isolated by collagenase digestion, washing, and separation. The results provide interesting information about the whole intrahepatic bile duct mass, and is expected to be interesting to readers.

7. *“Throughout the manuscript (particularly in the NIK knockout model) there is a lot of non-epithelial proliferation and apoptosis however you do not account for this? You recognise in the text that the relationship between BECs and the stroma is a complex one and likely interrelated, but fail to address clear changes in the microenvironment when BEC NIK is deleted”.*

We measured the proliferation of hepatocytes, HSCs, and Kupffer cells in DDC-fed mice, following these comments. Proliferating hepatocytes (Ki67⁺HNF4α⁺) and Kupffer cells (Ki67⁺F4/80⁺) were comparable both between *NIK*^{ΔK19} and *NIK*^{f/f} mice (Supplemental Fig. S4A) and between C33- and vehicle-treated mice (Supplemental Fig. S11B-C). HSC proliferation (Ki67⁺αSMA⁺) was decreased by C33 in DDC-fed mice (Supplemental Fig. S11B-C).

8. “You should characterise the cholangiocyte lines generated in SF1B more completely. K19 staining alone is insufficient”.

We measured the expression of additional cholangiocyte markers Cftr and Hnf1β. K19, Cftr and Hnf1β levels were higher in cholangiocyte lines (Supplemental Fig. S2B).

9. “It is not clear from the manuscript how TWEAK regulates NFκB2 - you should explain the relationship here more - why not treat with TNF”?

We expanded background discussion about TWEAK, following these comments. TWEAK has been reported to directly promote cholangiocyte proliferation and ductular reaction. NIK is required for TWEAK signaling; in contrast, NIK is dispensable for TNFα signaling.

10. “The authors describe changes in the immune microenvironment in *NIK* KO BECs. How does this work? what is the cytokine/chemokine profile of the BECs with and without *NIK* - it was recently reported that TWEAK induced MCP1 expression in cholangiocarcinoma, is this the same mechanisms here”?

The reviewer raised interesting questions. Respectfully, we argue that these questions should be thoroughly investigated in separate papers. It will take the enormous amount of time and efforts to identify and characterize cholangiocyte-secreted factors that influence liver inflammation and fibrosis. Nonetheless, we conducted the pilot studies to address these questions in cholangiocyte cultures. Indeed, deletion of *NIK* decreased the expression of MCP1 and other cytokines in TWEAK-stimulated cholangiocytes (Fig. for reviewer 3).

11. “the authors do not really investigate how *NIK* is playing a role in the microenvironment more generally - this should be explored. Is the reduction in fibroblasts/fibrosis/immune cells due to a reduction in epithelial cells or is this independent of epithelia number - if the latter, what is being produced by the WT epithelium that is stimulating the formation of the microenvironment”.

We argue, respectfully, that it would be more appropriate to study detailed molecular steps underpinning the interplays between cholangiocytes and other liver in separate papers (beyond the scope of this paper). Reactive cholangiocytes have been known to gain proinflammatory and profibrosis

properties. Biliary NIK increases the number of reactive cholangiocytes (ductular reaction), thus increasing secretion of cholangiocyte-derived mediators that promote liver inflammation and fibrosis. Additionally, cholangiocyte-intrinsic NIK may directly promote secretion of these mediators, further augmenting liver inflammation and fibrosis. We added these putative mechanisms to the revised discussion, following these comments. We wish to point out that it will take many years to identify and characterize the mediators. We also wish to point out that this paper uncovers, for the first time, the pivotal role of biliary NIK in the ductular reaction, liver injury, inflammation, and fibrosis. It provides proof of principle evidence that NIK inhibitors have promising translational potential. These findings are expected to be interesting to many investigators in the field of liver disease research.

12. *“What is the recombination efficiency of the K19Cre line and have the authors confirmed that in these cells there is always loss of both the IKK α allele and NIK allele at the same frequency”?*

We stained liver sections with antibodies to NIK and K19, or IKK α and K19, following these comments. *NIK* was deleted in most cholangiocytes in *NIK Δ K19* mice (Fig. 2A-B). Similarly, *IKK α* was deleted in most cholangiocytes in *IKK α Δ K19* mice (Supplemental Fig. S10A).

13. *“If NIK is not acting through IKK α , how is it acting and what is the mechanism here”?*

We conducted new experiments to explore Akt. Deletion of *NIK* decreased phosphorylation of Akt in cholangiocytes in response to TWEAK (Supplemental Fig. S5E). Pharmacological inhibition of the PI3K/Akt pathway, via wortmannin or LY294002, blocked the ability of TWEAK to stimulate the proliferation of *NIK $^{+/+}$* , but not *NIK $^{-/-}$* , cholangiocytes (new Fig. 3F). Thus, the PI 3-kinase/Akt pathway mediates, at least in part, the NIK action. We will further validate the biliary NIK/PI 3-kinase/Akt pathway in mice in the future (beyond the scope of this paper).

Reviewers' Comments:

Reviewer #1:

Remarks to the Author:

The authors have addressed my concerns.

Reviewer #2:

Remarks to the Author:

The authors have addressed some criticism but they failed to answer several major deficiencies of this study.

- The whole issue of link between NIK and Akt is not sufficiently examined (in response to criticism from Reviewers 1 and 3)
- The answers to Reviewer 2 regarding questions about how FBS activates NIK, and validation of NIK mediated degradation do not address the criticism. In addition, the quality of data and reagents is very questionable. For example, western blot for cIAP1 in supp fig 2H is of a very poor quality, and anti-mouse cIAP2 antibody that the authors used seems to be discontinued making it impossible to reproduce their data.
- Criticism regarding the use of C33 reagent is also not addressed.
- Importantly, comments from Reviewer 3 about small and insufficient number of human samples were not addressed (comment #1)
- Similarly, the authors introduce new data in response to comments under #2 from Reviewer 3 but they used only 4 mice per group – clearly insufficient.
- Comment #11 from Reviewer 3 is also not addressed.
-

Reviewer #3:

Remarks to the Author:

The revision presented here addresses the majority of my previous comments.

Given the level of journal, I do disagree with the authors on point 10 and 11 of their rebuttal and would have expected them to make more effort to include an analysis of the secreted factors that regulate the immune and stromal microenvironment in this system. This is a core tenant of this paper and for completeness I would have hoped to have this explored further.

This is particularly important given the published work on TWEAK/FN14 and the ductular reaction - this paper doesn't have huge levels of novelty and I continue to believe exploring the interesting, non-epithelial components of this system would be of benefit and take this manuscript beyond what is already known in the field.

As part of my previous review, I suggested that MCP1 was an interesting target and this came out in your preliminary experiments, even some more validation of this could be an important addition to the field and tie your phenotypes together.

Response to Reviewer #1:

"The authors have addressed my concerns".

We greatly appreciate Reviewer #1' constructive comments.

Response to Reviewer #2:

"The authors have addressed some criticism but they failed to answer several major deficiencies of this study.

- The whole issue of link between NIK and Akt is not sufficiently examined (in response to criticism from Reviewers 1 and 3)".

We appreciate these comments. Unfortunately, we currently do not have animal models to investigate the NIK/Akt pathway in vivo. We added this limitation in the revised discussion.

"- The answers to Reviewer 2 regarding questions about how FBS activates NIK, and validation of NIK mediated degradation do not address the criticism. In addition, the quality of data and reagents is very questionable. For example, western blot for cIAP1 in supp fig 2H is of a very poor quality, and anti-mouse cIAP2 antibody that the authors used seems to be discontinued making it impossible to reproduce their data".

FBS, which contains multiple NIK activators, was used to stimulate cholangiocyte proliferation in cell cultures, allowing us to determine the impact of NIK deficiency on cholangiocyte divisions. To complement this approach, we stimulated *NIK*^{+/+} and *NIK*^{-/-} cholangiocytes with TWEAK that has been known to activate NIK. In both FBS and TWEAK models, NIK deficiency decreases cholangiocyte proliferation. It might be distractive, in our view, by expanding the studies to delineate how FBS activates NIK in this context. Respectably, we believe that it is unlikely that detailed data illustrating molecular steps from FBS stimulation to NIK activation would provide an additional support for the conclusion about biliary NIK to stimulate ductular reaction.

To address the concerns about anti-cIAP2 antibody and data quality, we purchased new anti-cIAP1/2 antibodies from a different resource ABclonal. We repeated the experiments, improved data quality, added new Supplemental Fig. 2I (new antibodies from ABclonal), and replaced Supplemental Figs. 2H and 2J with new images, following these comments.

"- Criticism regarding the use of C33 reagent is also not addressed".

We guess that this comment may refer to *"This argues against safe and efficacious use of NIK inhibitor C33 (or other NIK inhibitors) in liver diseases given the pleiotropic nature of NIK function/activity"*. We provided new data showing that C33 treatment partially inhibited hepatic NIK under the experiments (Supplemental Fig. 11A). In contrast, biliary NIK was completely abated in *NIK*^{ΔK19} mice. Partial inhibition, rather than complete inactivation, of biliary NIK may explain the reduced effect of C33 compared to genetic deletion of biliary *NIK*. Nonetheless, C33 treatment still substantially alleviated DDC-induced liver injury (Fig. 8). It is expected that additional preclinical studies (optimizing C33 dose and treatment regimen) will improve C33 efficacy. Structure/activity relationship (SAR)-based optimization of C33 will lead to new NIK

inhibitors with improved efficacy (beyond the scope of this paper). It is not unusual that protein kinases have multiple molecular targets and pleiotropic functions, and this does not prevent them from serving as drug targets. Following these comments, we added limitations of NIK inhibitors to Discussion.

“- Importantly, comments from Reviewer 3 about small and insufficient number of human samples were not addressed (comment #1)”.

We added 4 human samples in response to the Reviewer 3's comments. Unfortunately, we are unable to obtain additional human liver biopsies. We added this limitation of human liver sample number to Discussion, following this comment.

“- Similarly, the authors introduce new data in response to comments under #2 from Reviewer 3 but they used only 4 mice per group – clearly insufficient”.

We increased mouse number from n=4 per group to n=6 (*NIK^{ΔK19}*) and n=7 (*NIK^{ff}*) in Figs. 2G, 3B, 4E, and 5D, following this comment.

“- Comment #11 from Reviewer 3 is also not addressed”.

We performed new experiments and added entirely new Fig. 7 to address the Reviewer 3's question (please refer to the Response to Reviewer 3).

Response to Reviewer #3:

“The revision presented here addresses the majority of my previous comments.

Given the level of journal, I do disagree with the authors on point 10 and 11 of their rebuttal and would have expected them to make more effort to include an analysis of the secreted factors that regulate the immune and stromal microenvironment in this system. This is a core tenant of this paper and for completeness I would have hoped to have this explored further.

This is particularly important given the published work on TWEAK/FN14 and the ductular reaction - this paper doesn't have huge levels of novelty and I continue to believe exploring the interesting, non-epithelial components of this system would be of benefit and take this manuscript beyond what is already known in the field.

As part of my previous review, I suggested that MCP1 was an interesting target and this came out in your preliminary experiments, even some more validation of this could be an important addition to the field and tie your phenotypes together”.

We greatly appreciate R#3' comments. We conducted additional experiments and added an entirely new Fig. 7 (the original Fig. 7 was renamed Fig. 8), following these comments. Ablation of NIK decreased expression of *Mcp1* as well as a battery of other cholangiokines and

mediators (Il1 β , Il4, Il6, iNos, Tnf α , Tgf β 1) in cholangiocyte cultures (Fig. 7A). To demonstrate that biliary NIK promotes secretion of inflammatory cholangiokines, we cocultured BMDMs with *NIK*^{-/-} or *NIK*^{+/+} cholangiocytes. Coculture with *NIK*^{+/+} cholangiocytes activated BMDMs to a higher level relative coculture with *NIK*^{-/-} cholangiocytes (Fig. 7B). To corroborate these results, we prepared *NIK*^{-/-} and *NIK*^{+/+} cholangiocyte-conditioned media and stimulated BMDMs with the media. *NIK*^{+/+} cholangiocyte-conditioned medium activated BMDMs to a higher level compared to *NIK*^{-/-} cholangiocyte-conditioned medium (Fig. 7C). We also stimulated HSCs with *NIK*^{-/-} or *NIK*^{+/+} cholangiocyte-conditioned medium. *NIK*^{+/+} cholangiocyte-conditioned medium stimulated HSC activation (liver fibrosis) to a higher level relative to *NIK*^{-/-} cholangiocyte-conditioned medium (Fig. 7D). We proposed a new model wherein biliary NIK stimulates secretion of cholangiokines, including Mcp1, Il1 β , tnf α , and Tgf β 1 (Fig. 7E). These cholangiokines act in concert, rather than alone, to stimulate liver immune cells and HSCs, leading to liver inflammation and fibrosis. NIK-elicited ductular reaction increases the NIK⁺ cholangiokine pool, further increasing secretion of cholangiocytes that exacerbate liver disease progression.

Reviewers' Comments:

Reviewer #2:

Remarks to the Author:

No additional comments.

Reviewer #3:

Remarks to the Author:

My concerns around the inflammatory phenotype have now been addressed in Figure 7. I have no further comments.

Response to Reviewer 2

Reviewer #2 (Remarks to the Author):

“No additional comments”.

Thanks.

Response to Reviewer 3

Reviewer #3 (Remarks to the Author):

“My concerns around the inflammatory phenotype have now been addressed in Figure 7. I have no further comments”.

Thanks.